# Microbes can help explain the evolution of host altruism

Ohad Lewin-Epstein[1,*], Ranit Aharonov[1,†,*] & Lilach Hadany[1]

The evolution of altruistic behaviour, which is costly to the donor but beneficial for the recipient, is among the most intriguing questions in evolutionary biology. Several theories have been proposed to explain it, including kin selection, group selection and reciprocity. Here we propose that microbes that manipulate their hosts to act altruistically could be favoured by selection, and may play a role in the widespread occurrence of altruism. Using computational models, we find that microbe-induced altruism can explain the evolution of host altruistic behaviour under wider conditions than host-centred theories, including in a fully mixed host population, without repeating interactions or individual recognition. Our results suggest that factors such as antibiotics that kill microbes might negatively affect cooperation in a wide range of organisms.

[1] Department of Molecular Biology and Ecology of Plants, Tel-Aviv University, Tel-Aviv 6997801, Israel. † Present address: IBM Research, Haifa 3490002, Israel. * These authors contributed equally to this work. Correspondence and requests for materials should be addressed to L.H. (email: lilach.hadany@gmail.com).

The evolution of altruism has been widely studied since Darwin's time[1–3]. Three major theories proposed to explain this phenomenon are: kin selection, proposing that natural selection can favour altruistic behaviour between kin[4,5]; reciprocity, which suggests repeating interactions[6] or individual recognition[7,8] as key factors; and group selection, which posits that altruism is favoured because of selection between groups[9]. All theories trying to explain the widespread occurrence of altruism have focused on the altruistic individual or its genes. Here we shift the focus from the individual performing an altruistic act to the microbes it hosts.

Almost any organism hosts microbes or other symbionts[10]. A growing body of evidence shows that microbes and symbionts can mediate behavioural changes in their hosts, in some cases improving their own fitness and transmission ability[11,12]. This has been shown in viruses (for example, rabies increasing aggression and contact[13]), macroparasites (for example, worms manipulating their cricket host to commit suicide[14]), plasmids (inducing their bacterial hosts to produce common goods[15]) and in particular bacteria[16,17]. More recently, it has been shown that the gut microbiome can affect the brain via the microbiome-gut-brain-axis[18–21]. Lactobacillus, for example, was shown to affect emotional behaviour in mice via the vagus nerve[22]. Recent evidence demonstrates that microbes are capable of manipulating the social behaviour of their hosts[23], and suggests that such manipulation has been subject to natural selection.

We propose that natural selection on microbes may favour manipulation of the host so that it acts altruistically, and that this may help explain the evolution of altruism in a wide range of hosts. Microbes can transfer horizontally from one host to another during host interactions. Following horizontal transfer, the recipient host may carry microbes that are closely related to the microbes of the donating host, even when the two hosts are unrelated. Microbes can also transfer vertically, from parent to offspring. As a result, a microbe that induces its host to help another host, increases the other host's survival or reproduction, thus increasing the vertical transmission (VT) of the microbes of the recipient host. Kin selection among the microbes could therefore favour microbes that induce altruistic behaviour in their hosts, thereby increasing the transmission of their microbial kin. We use population genetic models to investigate this hypothesis, and show that altruism induced by the host's microbes can spread in a population under much wider conditions than altruism coded by the host's own genes.

## Results

**Model description**. We consider a population of asexual individuals. We assume that each individual hosts one of two microbe types: microbes of type α manipulate their host to act altruistically, while microbes of type β have no effect on behaviour. Individuals interact in pairs, with a prisoner's dilemma payoff[24] (Fig. 1a): a host acting altruistically pays a fitness cost $1 > c > 0$, and the recipient gains a benefit $b > c$. During host interaction, microbes can be transmitted between the interacting hosts with probabilities $T_\alpha$ and $T_\beta$. $T_\alpha$ represents the probability of microbes of type α being transmitted to the other host, replacing the resident microbes, and likewise for $T_\beta$ (Fig. 1b). This direct link between interaction and the possibility for horizontal transmission is at the core of our model and differs from all related works[25,26]. At the end of each generation, individuals reproduce according to their fitness, microbes are vertically transmitted from one generation to the next, and the offspring generation replaces the parent generation.

We first investigate the special case where hosts behave altruistically only when carrying microbe α, there is no intrinsic

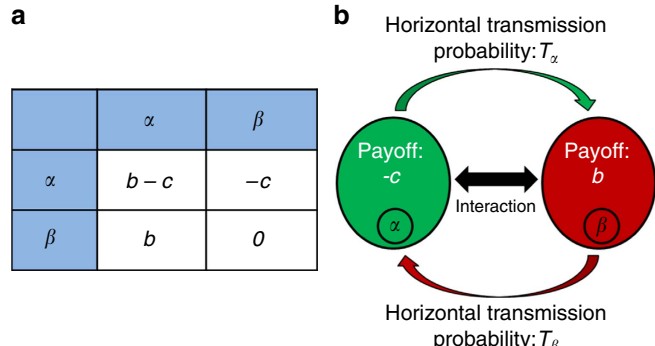

**Figure 1 | Interactions among pairs yield fitness change and a chance for horizontal transmission.** (**a**) Payoff matrix. An individual carrying microbes of type α acts altruistically: in each interaction it pays a fitness cost $c$, and its partner receives a fitness benefit $b$. Microbe β does not affect behaviour. (**b**) When two individuals interact, their fitness changes according to the payoff matrix. In addition, when the interacting individuals host different microbes, horizontal transmission may occur. With probability $T_\alpha$, microbe α is transmitted to the other host and establishes, replacing β. With probability $T_\beta$, microbe β is transmitted to the other host and establishes, replacing α. Transmission and establishment of one microbe is independent of the other microbe, and when both occur, they occur simultaneously.

cost to carrying a microbe, and offspring always inherit their parent's microbe (all three assumptions are relaxed below, yielding the same qualitative results). We compare the evolution of microbe-induced altruism with the classical case of altruism encoded genetically in the host, with perfect vertical transmission, no horizontal transmission, neglecting mutations, and using the same parameters $b$ and $c$.

**Fully mixed populations**. Consider an infinite, fully mixed population, that is, each individual has the same probability of interaction with any other individual in the population. Proportion $p$ of the individuals host microbe α, and proportion $q = 1 - p$ host microbe β. In each generation the population is randomly divided into pairs in which a single interaction occurs, with potential for microbe horizontal transmission (Fig. 1). After interaction, individuals reproduce according to their fitness, which is determined by the interactions they had. Microbe-induced altruism spreads when $p$, the proportion of hosts carrying microbe α, increases from one generation to the next. This happens when (see Methods):

$$T_\alpha b > c(1 - T_\beta) + (T_\beta - T_\alpha) \qquad (1)$$

Under equal horizontal transmission ($T_\alpha = T_\beta = T$), condition (1) reduces to a simpler form $\frac{b}{c} > \frac{1-T}{T}$. A more general condition for the increase in $p$ under relaxed model assumptions are detailed in the Methods, and further investigated in Supplementary Notes 1 and 2.

A few insights arise from condition (1). First, this condition does not depend on the proportion of altruists. This means that if (1) is satisfied, hosts carrying α will increase in proportion in the next generation, regardless of their current proportion in the population. That is, altruism will take over the population, even from rarity. Second, when $T_\alpha = 0$, condition (1) is never satisfied. That is, microbe-induced altruism cannot evolve in the absence of horizontal transmission of microbe α. Analogously, altruism encoded in the host genes, which also does not transmit horizontally, cannot evolve in such fully mixed populations (see Methods and previous works[2]). Third, condition (1) shows

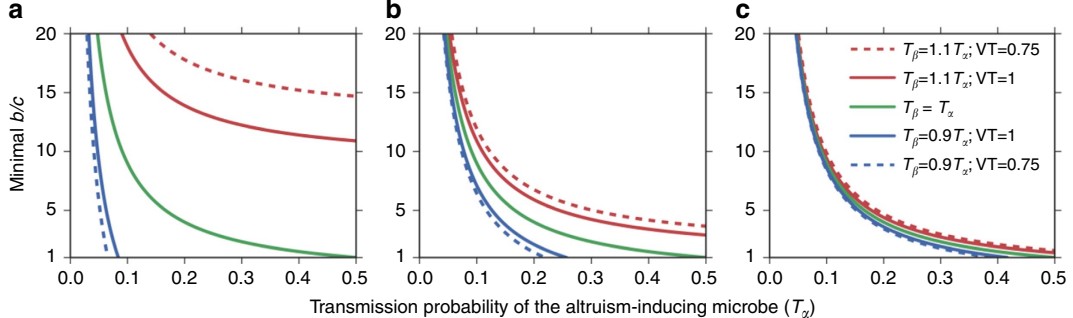

**Figure 2 | Horizontal transmission facilitates the fixation of altruism-inducing microbes.** Using condition (1) we calculate the minimal $b/c$ value that allows the fixation of microbe $\alpha$ for different values of the cost $c$ (subfigures **a**, **b**, **c**, with $c = 0.01$, $c = 0.05$ and $c = 0.2$, respectively), $T_\alpha$, $T_\beta$ and vertical transmission VT. For all $c$ and VT values, the critical $b/c$ value decreases with increasing horizontal transmission, even when $T_\alpha < T_\beta$ and vertical transmission is imperfect (VT<1). When the horizontal transmission probabilities are equal $T_\alpha = T_\beta = T$ (green solid lines), the condition for the spread of altruism becomes $\frac{b}{c} > \frac{1-T}{T}$, for any VT > 0 (see Methods for details). Thus, the line depends only on $T$ and is identical in all three subplots. However, the altruism-inducing bacteria spreads more slowly when VT<1 (Supplementary Note 1, Rate of $\alpha$'s spread as a function of vertical transmission). As $c$ increases (from **a** to **c**), the fitness effect of interaction on vertical transmission increases, diminishing the relative effect of imbalance between the horizontal transmission rates. The effect of imperfect vertical transmission (VT<1), is opposite, diminishing the effect of fitness differences on vertical transmission, thus giving more weight to imbalance between the horizontal transmission rates (compare red and blue solid lines to dashed lines). Presented are $b$, $c$ parameters within the range of the prisoner's dilemma (namely, $b > c$). All curves have an asymptote at $T_\alpha = 0$, namely altruism cannot evolve without horizontal transmission. Similarly, altruism cannot evolve in such a fully mixed population when it is encoded in the host's genome (see Methods).

resemblance to Hamilton's rule which considers the relatedness between donor and recipient, $r$, defined as the probability that two alleles drawn at random from the two individuals are identical by descent[27]. According to Hamilton's rule[4], altruistic behaviour towards kin is favoured if $r \cdot b > c$, that is, if the product of the benefit to the recipient, $b$, and the relatedness between donor and recipient, $r$, is greater than the cost to the donor, $c$. In the case of microbe-induced altruism, the spread from rarity of an altruism-inducing microbe can be described using the relatedness of the microbes of the two interacting hosts. While the identity of a host genotype is stable within a generation, the identity of its microbes may change: a rare altruism-inducing microbe can meet a relative (with probability zero in our fully mixed model) or infect the individual it meets and turn its microbes into relatives (with probability $T_\alpha$). Thus, with probability $T_\alpha$, manipulation by $\alpha$ microbes causes their host to help another host that now (after the interaction) carries relatives of the manipulating microbe $\alpha$. Furthermore, the altruism-inducing microbe may be replaced because of infection of its host by a different microbe with probability $T_\beta$, and in that case the cost paid by the host has no effect on the original microbe $\alpha$. Finally, the factor $(T_\beta - T_\alpha)$ represents the direct horizontal transmission disadvantage of $T_\alpha$ during interaction.

Solving condition (1) shows that the critical value of $b/c$ needed for the evolution of microbe-induced altruism decreases with increasing horizontal transmission probability (Fig. 2, solid lines). In other words, horizontal transmission of microbes helps the establishment of altruism in the host population. This is true even when the horizontal transmission probability of $\alpha$ is lower than that of $\beta$, corresponding to a within-host disadvantage for $\alpha$ (Fig. 2, solid red lines). When the horizontal transmission probability of $\alpha$ is higher, corresponding to the case that the altruistic behaviour increases the rate of transmission (for example, feeding), altruism evolves more easily (Fig. 2, solid blue lines).

Condition (1) was derived under the simplifying assumption of perfect vertical transmission. Relaxing this assumption, we generalized the model to assume imperfect vertical transmission of microbes, where with probability VT an offspring inherits

its parent's microbe, and with probability $1 - $ VT it inherits a random microbe from the parent population. We find that horizontal transmission facilitates the evolution of microbe-induced altruism even when vertical transmission is far from perfect (Fig. 2, dashed lines).

So far, we considered microbe-induced altruism in the absence of altruism induced by host genes. Extending our model, we consider a population that is polymorphic with respect to both altruism-inducing host genes and altruism-inducing microbes, and find that all our results hold: Altruism encoded genetically in the host does not evolve, irrespective of the presence of microbe-induced altruism, while the evolution of microbe-induced altruism is independent of the presence of altruism encoded in the host's genes (Supplementary Note 3). Our model is also robust to the addition of a baseline level of host altruism (Supplementary Figure 2).

**Spatially structured populations.** One key explanation for the evolution of altruism relies on the existence of spatial structure[28–31]. In classical studies, individuals interact only with neighbours, which are more likely to be related to them, and therefore altruists are more likely to interact with altruists. In addition, the probability of repeating interactions with the same individual increases significantly compared with a fully mixed population. Both characteristics generate a higher potential for benefit to altruists, and allow altruism encoded in an individual's genome to evolve under certain parameters. We thus used simulations to investigate whether microbe-induced altruism further widens the parameter range allowing the evolution of altruism in a spatially structured population, compared with classical altruism encoded in the host's genome. By studying spatial models, we extend our analysis to populations that are subject to drift, local interactions, local transmissions, and limited dispersal.

The spatial simulation consists of a 2D $100 \times 100$ lattice grid, where each site is inhabited by an individual host. Individuals carry either microbe $\alpha$, which drives them to behave altruistically, or microbe $\beta$, which does not. During a generation every individual initiates $K$ interactions, each with a neighbour

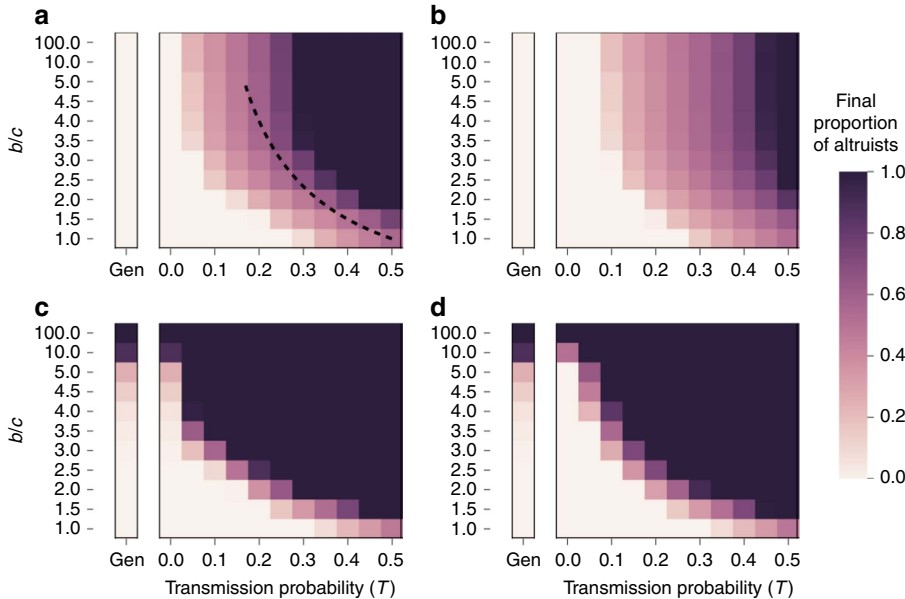

**Figure 3 | Microbe-induced altruism vs. altruism encoded in the host's genes in a spatial Prisoners' Dilemma scenario.** For microbe-induced altruism (matrix part of each sub-plot): hosts carrying either microbes of type α or β are placed on a 100 × 100 lattice grid. Hosts carrying microbe α initially inhabit 5% of the sites, chosen in random positions in the lattice. The final proportion of hosts that carry microbe α is plotted (colour-coded) as a function of horizontal transmission probability $T_\alpha = T_\beta = T$ and $b/c$ values, for (**a**) $K=1$, VT = 1, (**b**) $K=1$, VT = 0.75, (**c**) $K=8$, VT = 1 and (**d**) $K=8$, VT = 0.75, where $K$ is the number of interactions each host initiates per generation and VT is the vertical transmission of microbes. For altruism encoded in the host's genes (the first column in each plot, named 'Gen'): hosts carrying an allele for altruistic behaviour initially inhabit 5% of the sites, chosen at random. The final proportion of altruists is plotted (similarly colour-coded) as a function of $b/c$ for the same $K$ as described above. This is the classical case of altruism encoded genetically in the host, where vertical transmission is perfect, and no horizontal transmission occurs. Each cell in the plots represents the mean of at least 100 runs (see Methods for stopping criteria). For comparison with the analytic result of microbe-induced altruism, we plot in (**a**) the $b/c$ threshold derived from the analytical model, for the case of $K=1$ (the dashed line, plotted only in the range where the y scale is linear), as plotted in the green lines of Fig. 2a–c. As for the non-linear part of the y-axis, we get from the analytical model that for $T=0.1, 0.01, 0.001$ the critical $b/c$ values are 9, 99, and 999 respectively. For the case $T_\alpha \neq T_\beta$ we get very similar results (Supplementary Figure 4). We use $c=0.05$ throughout the simulation runs.

randomly chosen from its immediate neighbours (eight unless at the lattice edge; see Methods). To eliminate possible effects of the order of the interactions, each generation is divided into $K$ iterations over all individuals, where the order of the individuals initiating the interaction is randomized. The fitness of an individual is the sum of the payoffs it received from all its interactions according to the payoff matrix (Fig. 1), normalized by the number of actual interactions the individual had. In addition to fitness change, an interaction may also result in microbe horizontal transmission, with probabilities $T_\alpha$, $T_\beta$ as in the analytical model. Once all interactions are completed, reproduction takes place. Each site in the next generation grid is inhabited by a copy of the fittest individual in the neighbourhood consisting of this site and its immediate neighbours. The offspring inherits the microbe of its parent with probability VT. With probability $1 - VT$ it obtains the microbe of a randomly chosen individual in that neighbourhood (see Methods).

Our results show that similarly to the case of a fully mixed population, horizontal microbe transmission significantly extends the conditions allowing the evolution and maintenance of altruism. When individuals initiate one interaction per generation ($K=1$), microbe-induced altruism spreads in the population for a wide range of $b/c$ values, including a range of stable polymorphism (Fig. 3a). In contrast, altruism encoded in the host's genome does not persist even for high values of $b/c$ ('Gen' column in Fig. 3a). The parameter range allowing the evolution of altruism in the spatial model shows good agreement with the analytical results for a fully mixed population (see dashed line in Fig. 3a). Assuming that the vertical

transmission of microbes is imperfect (VT < 1) somewhat narrows the parameter range allowing the evolution of microbe-induced altruism (Fig. 3b), since it reduces the advantage of altruism-inducing microbes, which is based on enhancing the vertical transmission of the microbes in the recipient host. To compare with previous works that have shown that an allele for altruism can persist in a spatial model[30], we set the number of interactions per individual, $K$, to 8, and reset VT to 1. Indeed, for this case, altruism encoded in the host genes can persist for sufficiently high b/c values ('Gen' column in Fig. 3c), but the parameter range allowing persistence is wider for microbe-induced altruism, and widens with horizontal transmission probability ($T = T_\alpha = T_\beta$) (Fig. 3c). As in the case of a single interaction, imperfect vertical transmission has a mild effect on the parameter range allowing the evolution of microbe-induced altruism (Fig. 3d). Note that, as expected, when vertical transmission is perfect, microbe-induced altruism with zero horizontal transmission ($T=0$) is identical to the case of altruism encoded in the host genes (Fig. 3a,c), whereas for imperfect vertical transmission (VT < 1), this is not necessarily the case (Fig. 3d).

Finally, we tested if microbe-induced altruism can evolve from extreme rarity: we started with only a central $2 \times 2$ patch of individuals carrying microbe α while the rest of the population hosts microbe β. Figure 4 plots the proportion of runs in which $p$ reached 0.05 (complementing the analysis presented in Fig. 3a, where the starting proportion is 5%) for various parameters, and shows that microbe-induced altruism can increase in frequency even from extreme rarity, while altruism induced by the host genes cannot (see also Supplementary Videos 1 and 2) .

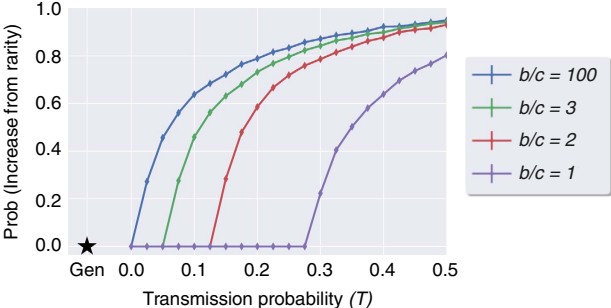

**Figure 4 | Horizontal transmission increases the probability of an extremely rare patch of an altruism-inducing microbe to increase in proportion in a spatial Prisoners' Dilemma scenario.** The estimated probability that a central $2 \times 2$ patch of hosts carrying microbe $\alpha$ will increase in proportion within a population of hosts carrying microbe $\beta$, is plotted as a function of $T = T_\alpha = T_\beta$, for several $b/c$ values, $K = 1$ and $VT = 1$. To estimate this probability, for each data point we ran at least 15,000 simulations until the proportion of $\alpha$ reached 0, 0.05 or stabilized in-between (stabilization below 0.05 happens in $< 10^{-4}$ of the runs, see Methods for stopping criteria). We then measured the proportion of simulations in which microbe $\alpha$ increased in proportion beyond 0.05—the initial proportion in Fig. 3a. This estimated probability that $\alpha$ will increase grows with $T$, and when $T = 0$ altruists do not increase from rarity for any $b/c$ value: the probability that $\alpha$ survives, when $T = 0$, was found not to be higher than 4/10,000 (the probability of a neutral microbe, identical in its effect on behaviour to microbe $\beta$, to fixate in such a model) based on 50,000 runs per $b/c$. The star ('Gen') represents the case of altruistic behaviour encoded in the host genome, where altruists do not increase from rarity for any $b/c$ value.

## Discussion

Our results – that microbes can facilitate the evolution of host altruism – imply a new perspective on various manifestations of altruistic behaviour. They may help illuminate intriguing cases of altruistic behaviour, including eusociality among social insects[32], especially with multiple queens and fathers, where relatedness is not very high[33]; mutual help between individuals from different species[34–36]; and animals caring for offspring of other parents or even other species[37]. Finally, our model shows for the first time that altruism can evolve even in well-mixed populations with neither repeating interactions nor individual recognition.

Our model can be considered in the context of classical theories for the evolution of altruism. It has been suggested that many of the previous models share a common principle[38]: that altruistic individuals preferentially help other altruistic individuals, according to kinship, memory, or group (see refs 39–42, but see for a different view refs 43,44). In our model the altruism-inducing microbe manipulates its host to help another host, irrespective of its microbes. Following the interaction, the receiving host may carry the relatives of the original microbe, and thus help is in effect preferentially directed towards future altruists. That is, the probability of helping someone that would be an altruist after the interaction $(p + qT_\alpha)$ is higher than the proportion of altruists in the general population $(p)$.

Our work can be extended in several important directions. First, our results suggest that a conflict might occur between host interests and microbe interests. Such a conflict can lead to a co-evolutionary arms race[45,46] with respect to altruistic behaviour, where the host evolves resistance to the altruism-inducing microbes, and the microbes evolve new ways of manipulating the host. Second, we have assumed here that hosts are incapable of identifying other hosts carrying similar

microbes, for example, by scent produced by the microbes[47]. If hosts are indeed more likely to interact with other hosts based on microbe similarity, this could further increase the positive effect of microbes on altruism. Third, more realistic modelling of the host microbiome could consider a diverse microbial population within a single host, where behaviour is determined by microbial composition.

Our results imply that factors that affect the microbiome (for example, antibiotics, probiotics, specific foods[48–52]) may have an effect on the altruistic behaviour of the hosts. In many cases the effect on altruistic behaviour could be an indirect result of an effect on other behaviours: for example reduction of social anxiety[22] may increase the probability of cooperative behaviour. Our results further suggest that the rate of microbe horizontal transmission could affect the evolution of altruism. We therefore predict that microbe-induced altruism is more likely to evolve when individuals interact in a way that easily allows horizontal transmission of microbes from one to the other, such as food sharing (vampire bats[53], offspring feeding by parent[54], trophallaxis among social insects nestmates[55]), but also touching, grooming and co-sheltering[56–58]. Our theoretical predictions call for experimental validation of whether microbes indeed mediate altruistic behaviour of their hosts, by what mechanisms, and whether elimination of microbes, for example, by antibiotics, hampers altruism.

## Methods

**The general microbe-induced altruism model.** In this section we describe the full model used for this study, in which vertical transmission is not necessarily perfect, there may be an intrinsic cost to carrying a microbe, and the hosts can have a non-zero level of altruism irrespective of the microbe they carry. This general model includes all the scenarios presented in the results section.

We assume that each individual hosts one of two microbe types. Microbes of type $\alpha$ manipulate their host to increase its altruistic behaviour, resulting in an additional cost of $c$ for the altruist, and an additional benefit $b$ for the recipient. Microbes of type $\beta$ have no effect on host behaviour. We also assume that all the individuals share a genetic background determining a baseline level of altruistic behaviour so that all individuals pay a fitness cost $c_g$ and receive a benefit $b_g$ when interacting. Thus a host that carries microbes of type $\alpha$ pays a fitness cost $c_g + c$, while a host that carries microbes of type $\beta$ pays only a fitness cost $c_g$. The interacting partner of a host that carries microbes of type $\alpha$ gains a benefit $b_g + b$, while the partner of a host that carries microbes of type $\beta$ gains only a benefit $b_g$. Individuals interact in pairs, with a prisoner's dilemma payoff: $0 \le b_g, b, c_g, c$, with $c < b$, $c_g < b_g$ and $c + c_g < 1$. Note that this formulation also covers the case where there is an intrinsic cost to carrying a microbe: since $c_g$ is uniform across the population, an equal cost for all microbe types can be introduced through an increase in $c_g$. Different costs to the different microbe types can be introduced by changing $c$ (and assuming the cost to carrying the microbe is applied before any horizontal transfer occurs).

Microbes are transmitted vertically, and in general this vertical transmission (VT) can be imperfect. With probability VT an offspring inherits its parent's microbe, and with probability $1 - VT$ it inherits a random microbe from the parent population. We use horizontal transmission probabilities $(T_\alpha, T_\beta)$, as defined in the results section. We define $p$ and $q = 1 - p$ to be the proportions of newborn hosts carrying microbes of type $\alpha$ and $\beta$, respectively, and $\hat{p}, \hat{q}$ to be the proportions of hosts carrying microbes of type $\alpha$, $\beta$, respectively, after interaction and before reproduction. Thus, if a parent carries microbes of type $\alpha$, with probability $(1 - VT) \cdot \hat{q}$ its offspring will carry microbes of type $\beta$, and if a parent carries microbes of type $\beta$, with probability $(1 - VT) \cdot \hat{p}$ its offspring will carry microbes of type $\alpha$.

Using the above, the proportion of hosts carrying microbes of type $\alpha$ after interactions including potential horizontal transmissions is:

$$\hat{p} = p^2 + pq(1 + T_\alpha - T_\beta) \qquad (2)$$

The mean fitness in the population in each generation is:

$$\bar{\omega} = (1 + b_g - c_g) + p(b - c) \qquad (3)$$

A newborn individual carrying microbes of type $\alpha$ is either an offspring of an individual that carried microbes of type $\alpha$ and transmitted its microbes vertically, or an offspring of an individual that did not transmit its microbes vertically to its offspring, who then got infected with microbes of type $\alpha$ (the latter can only happen when $VT < 1$). The proportion of individuals carrying microbes of

type $\alpha$ (altruists) in the next generation is therefore:

$$p' = \frac{1}{\bar{\omega}}[p^2(1+b_g+b-c_g-c)(1-(1-\mathrm{VT})\hat{q})$$
$$+pq(1-T_\beta)(1-c_g-c+b_g)(1-(1-\mathrm{VT})\hat{q})$$
$$+pqT_\alpha(1+b_g+b-c_g)(1-(1-\mathrm{VT})\hat{q})+pqT_\beta(1-c_g-c+b_g)(1-\mathrm{VT})\hat{p}$$
$$+pq(1-T_\alpha)(1+b_g+b-c_g)(1-\mathrm{VT})\hat{p}+q^2(1+b_g-c_g)(1-\mathrm{VT})\hat{p}]$$

(4)

We are interested in the case where altruism spreads in the population, that is, $p' > p$. This happens when:

$$p(1-p)[T_\alpha-T_\beta+T_\alpha b_g-T_\beta b_g-T_\alpha c_g+T_\beta c_g+\mathrm{VT}(T_\alpha b+T_\beta c-c)$$
$$+(1-\mathrm{VT})(T_\alpha bp-T_\beta bp-T_\alpha cp+T_\beta cp)] > 0$$

(5)

For $0 < p < 1$, we get:

$$T_\alpha-T_\beta+T_\alpha b_g-T_\beta b_g-T_\alpha c_g+T_\beta c_g+\mathrm{VT}(T_\alpha b+T_\beta c-c)+$$
$$(1-\mathrm{VT})(T_\alpha bp-T_\beta bp-T_\alpha cp+T_\beta cp) > 0$$

(6)

Whenever equation (6) is satisfied, we expect the proportion of $\alpha$ to increase from one generation to the next. Below and further in Supplementary Note 1, we analyse equation (6) under several parameter regimes.

**Derivation of condition (1).** If the vertical transmission is perfect (VT = 1), the condition for the spread of altruism, derived from equation (6), is:

$$T_\alpha-T_\beta+T_\alpha b_g-T_\beta b_g-T_\alpha c_g+T_\beta c_g+T_\alpha b+T_\beta c-c > 0 \quad (7)$$

Or, in a different formulation:

$$T_\alpha b > c(1-T_\beta)+(1+b_g-c_g)(T_\beta-T_\alpha) \quad (8)$$

When equation (8) is satisfied, the proportion of $\alpha$ will increase to fixation, since the condition does not depend on $p$.

If there is no genetic background of altruistic behaviour in the population ($b_g, c_g = 0$), then equation (8) reduces to condition (1) presented in the results section:

$$T_\alpha b > c(1-T_\beta)+(T_\beta-T_\alpha) \quad (9)$$

**Equal horizontal transmission probabilities.** If both microbes have the same horizontal transmission probability ($T_\alpha = T_\beta = T$), and there is no baseline altruism among the hosts ($b_g, c_g = 0$), the condition for the spread of altruism, derived from equation (6), becomes:

$$\mathrm{VT}(Tb+Tc-c) > 0 \quad (10)$$

Or (under the constraint that $\mathrm{VT} > 0$ and $T \neq 0$):

$$\frac{b}{c} > \frac{(1-T)}{T} \quad (11)$$

**No horizontal transmission.** When there is no horizontal transmission ($T_\alpha, T_\beta = 0$), no genetic background of altruistic behaviour in the population ($b_g, c_g = 0$), and perfect vertical transmission (VT = 1), equation (4) becomes:

$$p' = \frac{p^2(1+b-c)+pq(1-c)}{1+p(b-c)} \quad (12)$$

It is straightforward to see that for $p'$ as defined in equation (12) the condition $p' > p$ can never be satisfied (under the constraint $c > 0$), that is, altruism cannot evolve.

**Altruism determined by host genotype only.** We now analyse the classical dynamics of a population in which microbes do not affect altruistic behaviour, and the latter is fully determined by a locus with two alleles. Individuals that possess allele $A$ behave altruistically and pay a fitness cost $0 < c_g < 1$, whereas the recipient gets a fitness benefit, $b_g > c_g$. Individuals that possess allele $E$ do not pay a fitness cost or help their partner. We denote the proportion of individuals with allele $A$ by $p$, and those with allele $E$ by $q = 1 - p$. When deriving $p'$, the proportion of allele $A$ in the next generation we get:

$$p' = \frac{p^2(1+b_g-c_g)+pq(1-c_g)}{1+p(b_g-c_g)} \quad (13)$$

which is analogous to equation (12). As was the case for equation (12), condition (13) is never satisfied (under the constraint $c_g > 0$) and therefore altruism cannot evolve if encoded in the host genes.

**Simulation work flow.** For the case of microbe-induced altruism, a $100 \times 100$ lattice grid is formed where each site is inhabited by one host, carrying either microbes of type $\alpha$ or microbes of type $\beta$. In this simulation, individuals can interact only with their immediate neighbours. There are usually eight neighbours, unless the focal individual is close to one of the grid edges (Supplementary Fig. 5).

Each generation is composed of $K$ steps. At every step, each host in the lattice (drawn in a random order) interacts with a randomly drawn neighbour. During the interaction, hosts can pay a cost, $c$, and/or receive a benefit, $b$, according to their and their partner's behaviour, which is determined by the microbes they carry. In addition, in each interaction the microbes can be transmitted from one host to another, with probabilities $T_\alpha$, $T_\beta$ (transmission and establishment of one microbe is independent of the other microbe, and when both occur, they occur simultaneously). The fitness of each host is the sum of the payoffs it received from all its interactions according to the payoff matrix (Fig. 1), normalized by the number of interactions it participated in.

After all $K$ steps are over, reproduction takes place. Reproduction is modelled after Nowak and May[30]. A new lattice grid of the same size is formed. Every site in the new lattice is inhabited by a replicate of the fittest host from the same location, and its immediate neighbourhood, in the original lattice. If there are multiple hosts with the same maximal fitness in the neighbourhood, the parent is chosen at random from the fittest hosts (Supplementary Figure 6). In addition, if the vertical transmission is imperfect (VT < 1), then with probability $1 - \mathrm{VT}$, the offspring obtains the microbe of a randomly chosen individual from the neighbourhood of its location in the previous generation. $p$ denotes the proportion of hosts carrying $\alpha$ at that point in each generation.

For the case of altruism encoded in the host's genes, the simulation details are the same as for the microbe-induced altruism case described above, with the following differences: (1) The hosts do not carry microbes, they carry either an allele for altruistic behaviour ($A$) or an allele that does not affect behaviour ($E$), and $p$ denotes the proportion of allele $A$ in the population. (2) Vertical transmission is always perfect. (3) Because the hosts carry no microbes, no horizontal transmission takes place.

**Stopping criteria of the simulation.** For Fig. 3 the simulation was stopped when $p$, the proportion of microbe $\alpha$ (or the altruism-inducing allele $A$ in the case of altruism encoded in the host genes) reached 0 or 1, or when $p$ stabilized. Since $p$ may fluctuate, we measure stabilization by smoothing $p$ over a 200 generations window. We consider $p$ to be stable if the smoothed value of $p$ does not vary by $> 0.01$ in 200 generations. More formally, stabilization is calculated as follows: at each generation $k > 200$, $p_{smoothed}^k$ is calculated as the mean of $p$ in the previous 200 generations. The simulation is stopped at generation $g$ ($g > 400$) if:

$$\left|\max(p_{smoothed}^i) - \min(p_{smoothed}^j)\right| < 0.01 \text{ for } g - 200 < i, j < g \quad (14)$$

$p$ does not stabilize within 10,000 generations in $< 0.3\%$ of the runs, and for these runs, the measured $p$ is the one obtained from generation 10,000. For Fig. 4 the simulation is stopped when $p$ reaches 0 or 0.05, or when $p$ stabilizes, according to the above criterion (14).

**Data availability.** The simulation code is available from online service Zenodo, with doi: 10.5281/zenodo.192680.

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

## Acknowledgements

This project was supported by the Israel Science Foundation 1568/13 (L.H.). We thank Avigdor Eldar, Tuvik Beker, Judith Berman, Yoav Ram, Uzi Motro and Arnon Lotem for comments on the manuscript, and Eugene Rosenberg and Eyal Zinger for discussions.

## Author contributions

L.H. designed the study. O.L.-E., R.A. and L.H. formulated the model and derived the analytical equations. R.A. and O.L.-E. wrote the simulation code. R.A., O.L.-E. and L.H. analysed the results. O.L.-E. and R.A. contributed equally to the study. All authors discussed the results and took part in writing the manuscript.

## Additional information

**Competing financial interests:** The authors declare no competing financial interests.

