## [Peer Review File · Nature Communications]

Reviewers' comments:

Reviewer #1 (Remarks to the Author):

This manuscript uses a population genetic analytical model and simulations to investigate the evolution of altruism induced by symbiotic microbes in their hosts. In the model, all individuals carry a microbe; a proportion p of the host population carries microbe A, which induces cooperation in a prisoner's dilemma, and $1-p$ hosts carry microbe B, whose hosts defect. After an interaction in the prisoner's dilemma between partners carrying different microbes, microbe A displaces microbe B with a probability T_a , and B displaces A with probability T_b . ($T_a=T_b=0$ corresponds to the standard case without horizontal transmission of microbes.) The authors find the condition under which microbe A (and thus the altruistic trait) will spread in the population, and show that altruism cannot spread without horizontal transmission, and spread of altruism is not frequency-dependent. With horizontal transmission, altruism is more likely to spread when the T_a and T_b are higher, and when $T_a>T_b$. In a simulation of a structured population, the authors find that these results still hold, and additionally that altruism can spread when initially rare, and when hosts only interact with each other once.

I thought this was a fascinating and well written manuscript. The idea is intriguing, and the results are intuitively appealing, i.e. that the opportunity for horizontal transmission of an altruism trait allows its spread in otherwise unfavorable conditions. However, I have several comments on the manuscript that I think should be addressed before it is published. These can largely be addressed by explaining and justifying assumptions made in the model, and putting the model in its proper context within the literature on and natural history of cooperation.

Model set-up and assumptions:

In lines 51-79, the authors make a number of assumptions that are not fully acknowledged. I realize the necessity of making assumptions, but I think these limit the model's biological applicability, making the authors' claims of its utility sound overstated (see below). First, all individuals in the population carry a microbe (either A or B); second, there is no altruism unless induced by microbe A; third, there is no intrinsic cost to carrying a microbe (and thus it is not a parasite); fourth, hosts always transmit the microbe to their offspring (vertical transmission with probability); and fifth, there is no coevolution of behavior in the host encoded by its own genes.

In lines 89-91, I can see why we need to consider the relatedness between microbes, not hosts, but the costs and benefits are in terms of host fitness (line 54), so I'm unclear whose perspective is being taken here. The microbe's fitness obviously depends on its host's survival and reproduction, but cannot be the same as the host's fitness because only the microbe has opportunities for horizontal transmission.

In lines 91-95, I wasn't sure about the calculations of the post-interaction "relatedness".

Take a donor host harboring microbe A. With probability p , the recipient host will also have microbe A before the interaction (regardless of whether it is transmitted by the host). With probability $(1-p)T_a$, the recipient would have had microbe B before the interaction but received microbe A after the interaction. So the post-interaction probability that both hosts have microbe A would be $p+(1-p)T_a$, not T_a as stated in the text.

To calculate the relatedness of the actor's microbe before interaction to its own microbe after interaction: the actor will pick up microbe B from a recipient with probability $(1-p)T_b$, but will otherwise have microbe A, so the probability that it will have microbe A before and after the interaction is $1-(1-p)T_b$.

Perhaps my logic is mistaken, but I didn't follow how the authors got the values they did.

The authors seem to be assuming that the actor with the altruism microbe is only interacting with non-altruist recipients, but that is not stated explicitly.

For the simulations of spatial structure, I thought that not enough details were given to follow the set-up, and how it related to the first model. For example, why was this a simulation rather than an analytical model, as in the first part of the paper? Why was $p=0.5$ chosen as the starting point? (Were other values chosen too?) Did the authors investigate multiple interactions per generation in the first model, as they did in the simulation? Did the results for the "fully mixed" simulation with $K=1$ correspond to the results from the analytical model?

I liked the videos, but it would be informative to include the code for the simulation too.

Claims and context:

The key result is that when an altruism trait can be horizontally as well as vertically transmitted, it will spread more easily than when it can only be vertically transmitted. I think this is novel, but I found the authors' claims about when cooperation has been shown to evolve under only vertical transmission to be misleading, and their claims about the importance of their result in explaining cooperation in nature to be exaggerated.

For example, in lines 9-11, I don't think it's really accurate to say that "altruism is common in nature even under conditions that are not easily explained by [kin selection, group selection and reciprocity] - the authors don't give any examples of unexplained altruism, and ref 6 that is cited here does give an additional explanation (enforcement), which further leads the reader to think that the existing explanations for altruism cover most cases. (Note that much of ref 6 is concerned with mutualism and other cases of direct benefits, which are often not considered "true" altruism: see

e.g. West et al. 2007 J Evol Biol 20: 415-432.) Again, the examples in lines 159-162 are all well explained by current theory, and are not all altruism (e.g. cooperation between species often brings mutual direct benefits).

There also seems to be some confusion over population structure, leading to some misleading claims. I wasn't sure what the authors meant by a "fully mixed" population (e.g. lines 85-86, 113-119). A population can be fully mixed while having high relatedness and repeated interactions among individuals, which the authors imply is not the case. It is not true that this model is "the first time" that altruism evolves in a mixed population without repeated interactions (lines 18-19, 162-163) - for example, humans routinely exhibit altruism in these conditions!

Finally, although there may be kin selection among microbes for altruism (lines 46-47), if interacting hosts are unrelated, would the host genome select against altruism? I think this should be acknowledged (and if it is not a plausible outcome, explain why) - the literature on conflict between organelles' and nuclear genomes provides good context for this, but is not mentioned. It would be good to see empirical examples that show that the results are not just theoretically plausible but likely in nature, e.g. expanding on those in lines 167-169.

Additional minor comments are below:

Abstract and set-up (lines 1-79):

Line 7. Replace "- one that is costly to the donor... -" with "(costly to the donor...)"

8. Change "were" to "have been"

12-13, 31-32. The authors seem to be using "microbes" and "parasites" interchangeably (but see "microbes and parasites" in line 32), which is incorrect, especially in this context because carrying the microbe does not impose a cost on the host (other than the cost of altruism). I think "symbiont" is a more accurate term

20. Omit "direct"

21. Omit "theoretical" but say earlier that this manuscript is a model, perhaps in line 16: "We find in a population genetic model that..."

21. By "direct" do you mean "empirical"?

24. Replace "evolution" with "natural selection"

25. Clarify what kind of reciprocity: repeated interactions are necessary for direct reciprocity, but less so for indirect (A helps B, C helps A - no two players have to interact more than once) (although admittedly the latter is hard to evolve for other reasons)

- 28. Omit "propose to"
- 42. Replace "the microbes" with "parasites"
- 44. Replace "interaction" with "interactions between hosts"
- 48. Change "and show" to "to show"

Supplementary material for condition 1:

Change "equation 1" to "inequality 1"

What does "the analytical model" refer to?

First set of results (lines 80-112):

83, 103-104. How do you know that microbe A will reach fixation rather than spread to a stable polymorphism with microbe B?

98. Specify what Condition 1 is being solved for. Given that b/c is important for Figure 2, I think Condition 1 should be shown in the text rearranged to give the critical value of b/c .

100. Should be "a within-host disadvantage"

103. "Microbe" should be plural; should be "inequality 1", not equation

Figure 2. Why were these numerical values chosen? What is the possible range of b and c ? (For example, the values of c stated here are all <1 , but can c be >1 ?)

Spatial structure (lines 113-156):

114. Should be written "as opposed to interactions in a fully mixed population"

117. "interaction" should be plural

134. Replace "host" with "hosts"; "by that" with "thus"

142-143. I'm not sure what you mean by "at random order"

143. I'm not familiar with Moore neighborhoods - please could you explain?

145. "microbe switch" should be "microbes switching"

149. What does "randomly initializing the population at 50% A" mean? If you're setting the starting point at 50%, then it's not random.

149. Why are we assuming $T_a=T_b=T$ here, rather than $T_a\neq T_b$ as before?

Figures 3a and b: it would be good to say on the figure itself what the colors refer to, above the legend (otherwise not easy to see at a glance: buried in the caption).

Figure 3c. According to the caption, this figure shows "proportion of runs where A increased in frequency above 5% of the population", but I'm not sure how that corresponds to the y-axis label of "estimated survival probability"

Supplementary information on fig 3:

More explanation here would be helpful: I found this information hard to follow. For example, it would be useful to have a verbal explanation of when the simulation is stopped (my interpretation of what's given is that it's stopped when the proportion of A stops changing), and of how pksmoothed relates to what is shown in the text.

For fig 3c, why were there different numbers of runs for $T=0$ and $T>0$? (And why was the simulation not stopped according to the method for figs 3a and 3b?) What does "probability of A to survive" (rewrite as "survival probability of A") mean?

Conclusion (lines 157-177):

populations without recurring interactions" - this model is not "the first time"

165. Rephrase "Horizontal transmission of parasites could affect the evolution of altruism"

166. Omit "thus"; "microbes induced" should be "microbe-induced"

170. Change "shaking" to "disrupting"

171. Omit "in some cases" unless you say what these cases are

171. Rephrase: "We have assumed here that hosts are not capable of..."

173. Change "microbes similarity" to "microbe similarity"

174. "the effect of microbes on altruism" is quite vague: "positive effect" or "synergistic"?

175. Rephrase: "experimental validation of whether microbes indeed mediate..."

176-177. Rephrase: "whether elimination of microbes hampers altruism"

References:

Capitalize journal titles

Note that ref 1 is actually two separate books (Origin of Species, Descent of Man)

Reviewer #2 (Remarks to the Author):

The key results of this MS are the use of a population genetics model to show how the transmission of microbes (that cause their hosts to behave altruistically) among hosts that interact can increase overall cooperation among hosts. In their model hosts also pass their microbes onto their offspring. Overall I find the modeling of microbe induced cooperation to be new and interesting, but I do have some concerns about implementation, context setting, and conclusions.

One thing that might help the reader see the significance of this work would be to have some examples from nature (even if hypothetical) about how altruistic acts among hosts would benefit their microbes. It is easy to understand how increasing aggression (biting) in rabid mammals helps the rabies virus get to new hosts, but in this MS transmission to new hosts is built in to the model as one of the parameters. So in what additional ways might altruism among hosts help microbes?

It seems that every paper about altruism starts by stating what an unsolved problem altruism is in evolutionary biology, but the truth is that the theories mentioned by the authors: Kin selection, group selection, and reciprocal altruism have all been successful in explaining this phenomena. And they have all had many refinements since their introductory papers cited in the MS. The big controversy is not whether they work, but which is to be preferred in terms of simplicity or causal explanations-most of which really comes down to a preference for accounting methods. Their underlying similar mechanism is helping behaviors that are disproportionately directed towards the focal cooperative genotype. This assortment can be caused by conditional behavior (e.g. Tit-For-Tat), population viscosity, kin recognition, etc. The underlying similarities in these theories have been noted by many authors over the years, e.g., Breden 1990 *Evolution*, Wilson & Dugatkin 1997 *Am Nat*, Sober & Wilson 1998, Frank 1998, Page & Nowak 2002 *JTB*, Fletcher & Zwick 2006 *Am Nat*, although there has also been more recent criticisms of this equivalence viewpoint, e.g., van Veelen 2005 *JTB* and van Veelen et al 2012 *JTB*. The controversy about which theory is to be preferred should not be confused with failure of the theories to explain altruistic phenomena.

So I think it is too much to suggest that "microbes explain the evolution of host altruism", as if there is a void left by the failure of other theories that can be filled by microbe-induced altruism. I think it would make more sense to say how microbe mediated altruism fits into the framework of more cooperative behaviors being directed to altruists. There is some attempt to connect to Hamilton's rule, but here there is the assumption that high relatedness automatically leads to altruism. The r

term in Hamilton's rule is not simple genealogical relatedness, but rather a measure of above average covariance between the focal genotype and help from others.

I also think the assumptions in the model are rather extreme. It is assumed that microbes have total control over their host's behavior, whereas in reality the influence of microbes on hosts is likely to be weak, especially if hosts can evolve defenses to having their behavior altered in ways that lower their fitness.

My biggest problem with the MS is that there does not seem to be any genetic basis in the model for hosts to be cooperators (independent of their microbes). I can understand how zero transmission could be used as a proxy for host genetics in the well mixed population, but this is confusing and at a minimum should be explained more clearly. Also, while strong altruism may not evolve in well-mixed populations (without conditional behavior), it can come close to evolving if $b \gg c$. So microbial induced cooperation might push host cooperation to evolve when it is already close to evolving, but this enhancement of host altruism cannot be shown by the model if there is no host genetics for altruism. In other words, having an explicit trait for host altruism plus microbe caused altruism may give different results than assuming host altruism is zero and only looking at microbe caused altruism. Is there enhancement of host-mediated altruism by microbe-mediated altruism? The MS seems to imply that there is: L17-18 "microbe-induced altruism can explain the evolution of altruism under wider conditions than host-centered theories" and L124-135 "transmission significantly extends the conditions allowing for the evolution and maintenance of altruism. But this cannot be assessed in the model if there is no implementation of host-induced altruism.

Again, I think the idea of altruism being enhanced by microbes with their horizontal transfer is an interesting idea, but I don't think the current model, which seems to lack any explicit host genetics for altruistic traits, fully addresses this question. I also think some tie to nature would be very helpful. Under what circumstances might microbe-induced altruism between hosts be helpful to the microbes?

Specific Comments:

=====

L10-11: again, I don't agree that these theories have failed

L40: again, natural selection would also favor hosts that could resist having their behaviors altered in ways that hurt their fitness.

L49: "altruism coded by the host's own genes" implies that there are such genes in the model, but if they are in the model they are not explained.

L60-61: again "altruism encoded genetically in the host" suggest that such genes exist in the model.

L117-119: The idea of having repeated interactions (PD games) is that behavior in current interaction can be conditional on past interactions, e.g. Tit-For-Tat minimizes exploitation by defectors while gaining (mutual) benefits when interacting with other cooperative strategies. What is the point of repeated interactions in this model, if behavior is not conditional?

L125: why starting with such a high percent of alpha hosts?

L127: "As shown before" sounds like something discussed earlier in the MS, but I think authors are referring to other work in the literature. It might be clearer to say "As shown elsewhere" instead.

L129-132: This sentence sounds as if it is a finding from the model, but again where is the host altruism genetics that could support this statement?

Fig. 3: In both a) and b) on the bottom row when $b/c = 1$ there are stable polymorphisms, but one would expect cooperators to go extinct when $b = c$ because even if $r = 1$, Hamilton's inequality is not satisfied. Can you explain what is going on here?

L151-153: This description of c) does not seem to match the legend on the y-axis which says "survival probability"

L159-162: again, I don't think these examples are hard to explain using current theory.

Reviewers' comments:

Reviewer #1 (Remarks to the Author):

This manuscript uses a population genetic analytical model and simulations to investigate the evolution of altruism induced by symbiotic microbes in their hosts. In the model, all individuals carry a microbe; a proportion p of the host population carries microbe A, which induces cooperation in a prisoner's dilemma, and $1-p$ hosts carry microbe B, whose hosts defect. After an interaction in the prisoner's dilemma between partners carrying different microbes, microbe A displaces microbe B with a probability T_a , and B displaces A with probability T_b . ($T_a=T_b=0$ corresponds to the standard case without horizontal transmission of microbes.) The authors find the condition under which microbe A (and thus the altruistic trait) will spread in the population, and show that altruism cannot spread without horizontal transmission, and spread of altruism is not frequency-dependent. With horizontal transmission, altruism is more likely to spread when the T_a and T_b are higher, and when $T_a > T_b$. In a simulation of a structured population, the authors find that these results still hold, and additionally that altruism can spread when initially rare, and when hosts only interact with each other once.

I thought this was a fascinating and well written manuscript. The idea is intriguing, and the results are intuitively appealing, i.e. that the opportunity for horizontal transmission of an altruism trait allows its spread in otherwise unfavorable conditions. However, I have several comments on the manuscript that I think should be addressed before it is published. These can largely be addressed by explaining and justifying assumptions made in the model, and putting the model in its proper context within the literature on and natural history of cooperation.

Model set-up and assumptions:

In lines 51-79, the authors make a number of assumptions that are not fully

acknowledged. I realize the necessity of making assumptions, but I think these limit the model's biological applicability, making the authors' claims of its utility sound overstated (see below).

First, all individuals in the population carry a microbe (either A or B);

We indeed did not sufficiently clarify all of our assumptions. We now clarify this assumption on L55-57: “We consider a population of asexual individuals. We assume that each individual hosts one of two microbe types: microbes of type α manipulate their host to act altruistically, while microbes of type β have no effect on behavior.”

second, there is no altruism unless induced by microbe A;

We thank the reviewer for pointing this out and helping us generalize the model we present. In our revised manuscript we relax this assumption in two ways:

- a. We allow both host altruism and microbe induced altruism in the same population. We solve this generalized 4-types model (see SI section 1.5), and find that the results are robust to the addition of host altruism: host altruism never succeeds in this fully mixed model, and microbe-induced altruism evolves under the exact same conditions, irrespective of the presence of host altruism. See L149-156 and Fig. S3.
- b. We also model another case, where a certain level of host altruism is fixed prior to the appearance of the microbe. We find that our main results are robust to that assumption as well (SI 1.1-1.2)

third, there is no intrinsic cost to carrying a microbe (and thus it is not a parasite);

Following this comment, we now acknowledge the no-cost assumption in the description of the model (L68), and include a generalization relaxing this assumption (L69, L154-156; details in SI 1.1-1.3;)

fourth, hosts always transmit the microbe to their offspring (vertical transmission with probability);

Again, we thank the reviewer for this comment which directed us to a more general model. In the revised manuscript we relax this assumption and generalize the model to include imperfect transmission of the microbe between parent and offspring. We present the effect of imperfect vertical transmission both in the analytical model (revised Fig 2, dashed lines; pL131-136; SI 1.1-1.3) and in the spatial model (revised Figure 3b,d; L182-184), and discuss it L193-204. We show that the results are robust to imperfect vertical transmission. We think that this generalization has significantly strengthened our model. Thanks for the suggestion.

and fifth, there is no coevolution of behavior in the host encoded by its own genes. We acknowledge that this is a very interesting and important direction, but we found co-evolution to be beyond the scope of this paper. Following the reviewer's comment, we now acknowledge this important point explicitly in the discussion (L258-260).

In lines 89-91, I can see why we need to consider the relatedness between microbes, not hosts, but the costs and benefits are in terms of host fitness (line 54), so I'm unclear whose perspective is being taken here. The microbe's fitness obviously depends on its host's survival and reproduction, but cannot be the same as the host's fitness because only the microbe has opportunities for horizontal transmission.

This point was indeed not clearly explained in our original manuscript. It is true that the fitness of the microbe is different from that of the host. Inequality 1 describes a condition on microbe fitness, that includes b and c , since the vertical transmission of the microbes depends on the host fitness, but also T_α and T_β , which represents the microbes' horizontal transmission ability.

To better address this, in the revised manuscript we

- a. Clarified our presentation of the intuition for the model (L44-51) and in particular for condition 1: L109-120.
- b. Added a section to the SI (section 1.4) in which we derived the fitnesses of the two microbes ($\omega_\alpha, \omega_\beta$) explicitly, and show that analyzing the conditions in which $\omega_\alpha > \omega_\beta$ yields the same inequality as presented in the main text.

In lines 91-95, I wasn't sure about the calculations of the post-interaction "relatedness".

Take a donor host harboring microbe A. With probability p , the recipient host will also have microbe A before the interaction (regardless of whether it is transmitted by the host). With probability $(1-p)T_a$, the recipient would have had microbe B before the interaction but received microbe A after the interaction. So the post-interaction probability that both hosts have microbe A would be $p+(1-p)T_a$, not T_a as stated in the text.

To calculate the relatedness of the actor's microbe before interaction to its own microbe after interaction: the actor will pick up microbe B from a recipient with probability $(1-p)T_b$, but will otherwise have microbe A, so the probability that it will have microbe A before and after the interaction is $1-(1-p)T_b$.

Perhaps my logic is mistaken, but I didn't follow how the authors got the values they did. The authors seem to be assuming that the actor with the altruism microbe is only interacting with non-altruist recipients, but that is not stated explicitly.

We acknowledge that our presentation of this rather complicated matter was not clear. We thus revised our description of the relatedness issue.

Following the reviewer's comment, we added to our revised supplementary a section (1.4) in which we derive the expected fitness of each microbe based on the probabilities of the different possible interactions and transmission. We show that this analysis also results in condition 1.

In addition we extended the explanation of the intuition for the derivation in the revised text (L109-120), where we clarify that in our extreme "fully mixed" model, the probability that two interacting individuals carry two microbes that are identical by descent is assumed to be zero. We also clearly note that the explanation in the main text is only an intuition, while condition 1 itself is derived explicitly in the supplementary.

For the simulations of spatial structure, I thought that not enough details were given to follow the set-up, and how it related to the first model.

Following this comment, we rewrote the description of the simulations with much more detail, and added a new section in the SI (section 2), describing in detail the simulation setup and the stop criteria.

For example, why was this a simulation rather than an analytical model, as in the first part of the paper?

Spatial structure has been shown to affect the evolution of cooperation in many classical works (refs 29, 31, 32), and is common in natural populations. Thus we believe that considering the case of spatial structure is important for the comprehensiveness of our results. However, to add the realistic assumptions of spatial structure, finite population, local interactions from generation to generation, and local transmission, requires a dramatically more complicated model. Hence we (and the previous works studying similar models) used stochastic simulations to investigate.

Why was $p=0.5$ chosen as the starting point? (Were other values chosen too?)

Following the reviewer's comment, we investigated the dynamics starting from 5% rather than 50%, and the results are nearly unchanged. The results presented in the revised Fig. 3 start at 5% complementing the results in the revised figure 4 (ending at 5%). We believe presenting results starting at a low fraction of microbe-inducing microbes strengthens our work. Thanks for the comment!

Did the authors investigate multiple interactions per generation in the first model, as they did in the simulation?

We did not. In fact, the results with 8 interactions were mainly for comparison with the work by Nowak et al (Nowak & May, 1998) which describes the effect of spatial structure on altruism encoded in the host's genes (this was a simulation model as well). In addition, as was shown in (Nowak & May, 1998) and also in our results, gene-encoded altruism can be maintained in a spatial structured population when $K = 8$ (Fig. 3c, left column labeled 'Gen'). Our results show that even when gene-encoded altruism can be maintained, microbe-induced altruism can be maintained or even fixate under lower b/c values – namely more easily.

We now show the case of 8 interactions when starting from 5% (Fig. 3c, almost unchanged) and with imperfect vertical transmission (Fig. 3d). Having said that, we would be happy to move the

$K = 8$ results to the supplementary and concentrate on one interaction per generation in the text if the reviewers consider it clearer.

Did the results for the "fully mixed" simulation with $K=1$ correspond to the results from the analytical model?

Yes. This is indeed a very good point we neglected to show in our original manuscript. Following the reviewer's comment we added a direct comparison of the analytical and simulation models (dashed line in Fig. 3a, showing good agreement). Thanks!

I liked the videos, but it would be informative to include the code for the simulation too. The code will be uploaded to dryad upon acceptance

Claims and context:

The key result is that when an altruism trait can be horizontally as well as vertically transmitted, it will spread more easily than when it can only be vertically transmitted. I think this is novel, but I found the authors' claims about when cooperation has been shown to evolve under only vertical transmission to be misleading, and their claims about the importance of their result in explaining cooperation in nature to be exaggerated.

For example, in lines 9-11, I don't think it's really accurate to say that "altruism is common in nature even under conditions that are not easily explained by [kin selection, group selection and reciprocity] - the authors don't give any examples of unexplained altruism, and ref 6 that is cited here does give an additional explanation (enforcement), which further leads the reader to think that the existing explanations for altruism cover most cases. (Note that much of ref 6 is concerned with mutualism and other cases of direct benefits, which are often not considered "true" altruism: see e.g. West et al. 2007 J Evol Biol 20: 415-432.) Again, the examples in lines 159-162 are all well explained by current theory, and are not all altruism (e.g. cooperation between species often brings mutual direct benefits).

Following this comment, we rewrote the part referring to the classical theories of the evolution of altruism, trying to write a more balanced description, and not describing it as an unsolved problem (L7-9, 250-256). We also added a few references.

There also seems to be some confusion over population structure, leading to some misleading claims. I wasn't sure what the authors meant by a "fully mixed" population (e.g. lines 85-86, 113-119). A population can be fully mixed while having high relatedness and repeated interactions among individuals, which the authors imply is not the case.

By fully mixed population we mean a population in which "each individual has the same probability to interact with any other individual in the population" (L82-83). Therefore, when the number of individuals in the population is much larger than the number of interactions each individual undergoes, the probability to interact with the same individual more than once is very low. We clarify this in the text (L112-114): "...a rare altruism-inducing microbe can meet a relative (with probability zero in our fully mixed model)...".

It is not true that this model is "the first time" that altruism evolves in a mixed population without repeated interactions (lines 18-19, 162-163) - for example, humans routinely exhibit altruism in these conditions!

Thanks for pointing this out. Indeed, our statement wasn't accurate. We now refined this statement: "...shows for the first time that altruism can evolve even in well-mixed populations with neither recurring interactions nor individual recognition". This statement is correct to the best of our knowledge.

Finally, although there may be kin selection among microbes for altruism (lines 46-47), if interacting hosts are unrelated, would the host genome select against altruism? I think this should be acknowledged (and if it is not a plausible outcome, explain why) - the literature on conflict between organelles' and nuclear genomes provides good context for this, but is not mentioned.

Acknowledged with a few refs regarding co-evolutionary arms race, and see our answer re co-evolution above (L258-260).

It would be good to see empirical examples that show that the results are not just theoretically plausible but likely in nature, e.g. expanding on those in lines 167-169. We thank the reviewer for this comment. We now expanded the biological context in L31-40 and L267-275 by adding about 14 new refs and explaining the biological relevance in more detail.

Additional minor comments are below:

Abstract and set-up (lines 1-79):

Line 7. Replace "- one that is costly to the donor... -" with "(costly to the donor...)"

Done

8. Change "were" to "have been"

Done

12-13, 31-32. The authors seem to be using "microbes" and "parasites" interchangeably (but see "microbes and parasites" in line 32), which is incorrect, especially in this context because carrying the microbe does not impose a cost on the host (other than the cost of altruism). I think "symbiont" is a more accurate term

Changed. Thanks!

20. Omit "direct"

Done

21. Omit "theoretical" but say earlier that this manuscript is a model, perhaps in line 16: "We find in a population genetic model that..."

We removed “theoretical” and added “Using computational models, we find that microbe-induced altruism can explain the evolution of altruism”.

21. By "direct" do you mean "empirical"?

Indeed, we changed to empirical. Thanks!

24. Replace "evolution" with "natural selection"

Done

25. Clarify what kind of reciprocity: repeated interactions are necessary for direct reciprocity, but less so for indirect (A helps B, C helps A - no two players have to interact more than once) (although admittedly the latter is hard to evolve for other reasons)

We now refer to both types of reciprocity, with references. Thanks!

28. Omit "propose to"

Done

42. Replace "the microbes" with "parasites"

Following the reviewer’s previous comments, we removed “parasites” and we now refer only to “microbes” and “symbionts” throughout the manuscript.

44. Replace "interaction" with "interactions between hosts"

Done

48. Change "and show" to "to show"

Done

Supplementary material for condition 1:

Change "equation 1" to "inequality 1"

Done

What does "the analytical model" refer to?

We completely rewrote the supplementary and the analytical model is now referred to more clearly.

First set of results (lines 80-112):

83, 103-104. How do you know that microbe A will reach fixation rather than spread to a stable polymorphism with microbe B?

When deriving the condition for the spread of altruism (now better detailed in the new SI 1) we obtain inequality (1) that does not depend on the frequency of (p). This means that if condition (1) is satisfied, for all p values, p' will be greater than p , namely α will increase in proportion. We verified the fixation of the altruist inducing microbe using numerical simulation. To improve our presentation, we added a clarifying sentence (L99-101): "This means that if (1) is satisfied, α will increase in proportion in the next generation, regardless of its current proportion in the population. That is, altruism will take over the population, even from rarity." Note that under the more general condition we developed in the new SI, when $VT < 1$ there exists a very narrow range of parameters allowing polymorphism (see SI section 1.2). However, no polymorphism exists in the case of perfect vertical transmission presented in the main text.

98. Specify what Condition 1 is being solved for. Given that b/c is important for Figure 2, I think Condition 1 should be shown in the text rearranged to give the critical value of b/c .

We considered both presentations of the condition:

a. $T_{\alpha}b > c(1 - T_{\beta}) - (T_{\alpha} - T_{\beta})$

b. $\frac{b}{c} > \frac{(1-T_{\beta})}{T_{\alpha}} - \frac{(T_{\alpha}-T_{\beta})}{T_{\alpha}c}$

Admittedly, we debated about this quite a bit since each of these presentations has its pros and cons. For the analysis of the results, and in order to be consistent with classical studies, we investigate the b/c ratio needed for the evolution of altruism, and in this perspective, indeed the second option is better, as pointed out by the reviewer. However, the main advantage of the first option is that it is simpler to read and in addition it is easier to provide intuition for the condition, when presented in the first format.

We therefore consulted several readers that much preferred the simpler format of the current condition (1), and decided to choose that presentation. If the editor or the reviewer see it as critical, we are willing to change the format.

100. Should be "a within-host disadvantage"

Done

103. "Microbe" should be plural; should be "inequality 1", not equation

Done

Figure 2. Why were these numerical values chosen? What is the possible range of b and c ? (For example, the values of c stated here are all <1 , but can c be >1 ?)

The numerical values were chosen to be within the range of the prisoner's dilemma. We now clarify that and detail the range of b and c when presenting them (L58-59) "a host acting altruistically pays a fitness cost $0 < c < 1$, and the recipient gains a benefit $b > c$. "

Three values of c where chosen in order to show results for low, intermediate and high fitness cost.

Spatial structure (lines 113-156):

114. Should be written "as opposed to interactions in a fully mixed population"

Done

117. "interaction" should be plural

Done

134. Replace "host" with "hosts" Done; "by that" with "thus" section removed

142-143. I'm not sure what you mean by "at random order"

This was indeed written in a somewhat confusing way. We changed the simulation description and explained it differently. It now reads (L174-176): "In order to eliminate possible effects of the order of the interactions, each generation is divided into K iterations over all individuals, where the order of the individuals initiating the interaction is randomized."

143. I'm not familiar with Moore neighborhoods - please could you explain?

We omitted this term, and gave an alternative description

145. "microbe switch" should be "microbes switching"

Simulation description is now in the main text, and not in the figure text. Phrasing was changed.

149. What does "randomly initializing the population at 50% A" mean? If you're setting the starting point at 50%, then it's not random.

We meant that 50% of the sites are inhabited by altruists, but the positions of these sites, within the lattice, are chosen at random. This was indeed not written clearly enough. We now start with 5% rather than 50%, and hopefully better explain this in the figure description (L208-209): "Hosts carrying microbe α initially inhabit 5% of the sites, chosen in random positions in the lattice."

149. Why are we assuming $T_a=T_b=T$ here, rather than $T_a \neq T_b$ as before?

Differences between equal and non-equal horizontal transmission are less significant in the spatial simulation, and therefore we focused on the case of $T_\alpha = T_\beta$. Nevertheless, following the reviewer's comment we added simulation results for $T_\alpha \neq T_\beta$ as well to SI, section 2.3, and refer to these in the caption of Fig. 3.

Figures 3a and b: it would be good to say on the figure itself what the colors refer to,

above the legend (otherwise not easy to see at a glance: buried in the caption).

Indeed - Added to the figure 3.

Figure 3c. According to the caption, this figure shows "proportion of runs where A increased in frequency above 5% of the population", but I'm not sure how that corresponds to the y-axis label of "estimated survival probability"

We thank the reviewer for pointing out that this is confusing. Following this comment, we clarified the presentation of this figure. The y-axis label now reads "prob (increase from rarity)", and is explained in the caption (L229-236).

Supplementary information on fig 3:

More explanation here would be helpful: I found this information hard to follow. For example, it would be useful to have a verbal explanation of when the simulation is stopped (my interpretation of what's given is that it's stopped when the proportion of A stops changing), and of how pksmoothed relates to what is shown in the text.

We added a simulation setup to the supplementary information (SI, section 2.1). In addition, we now better explain the stopping criteria for the simulations results presented in figures 3 and 4 (SI, section 2.2).

For fig 3c, why were there different numbers of runs for $T=0$ and $T>0$?

We were more conservative with $T = 0$, in order to not only estimate the probability but also validate that it is not higher than a neutral mutation. We now explain this in the new Fig. 4 caption (L237-239).

(And why was the simulation not stopped according to the method for figs 3a and 3b?)

The simulation was in fact stopped by a nearly identical criterion, apart from the threshold (0.05 rather than 1). We now explain this in the caption (L234-235) and in the supplementary information (SI, section 2.2).

What does "probability of A to survive" (rewrite as "survival probability of A") mean?

The y-axis label now reads “prob (increase from rarity)”, and is explained in the caption (L230-236).

Conclusion (lines 157-177):

populations without recurring interactions" - this model is not "the first time"

This is true. We refined this statement: “our model shows for the first time that altruism can evolve even in well-mixed populations with neither recurring interactions nor individual recognition.” (L247-248)

165. Rephrase "Horizontal transmission of parasites could affect the evolution of altruism"

Done

166. Omit "thus"; "microbes induced" should be "microbe-induced"

Done

170. Change "shaking" to "disrupting"

Done

171. Omit "in some cases" unless you say what these cases are

Done

171. Rephrase: "We have assumed here that hosts are not capable of..."

Done

173. Change "microbes similarity" to "microbe similarity"

Done

174. "the effect of microbes on altruism" is quite vague: "positive effect" or "synergistic"?

Changed to “positive effect”. Thanks!

175. Rephrase: "experimental validation of whether microbes indeed mediate..."

Done

176-177. Rephrase: "whether elimination of microbes hampers altruism"

Done

References:

Capitalize journal titles

Done

Note that ref 1 is actually two separate books (Origin of Species, Descent of Man)

Fixed

Reviewer #2 (Remarks to the Author):

The key results of this MS are the use of a population genetics model to show how the transmission of microbes (that cause their hosts to behave altruistically) among hosts that interact can increase overall cooperation among hosts. In their model hosts also pass their microbes onto their offspring. Overall I find the modeling of microbe induced cooperation to be new and interesting, but I do have some concerns about implementation, context setting, and conclusions.

One thing that might help the reader see the significance of this work would be to have some examples from nature (even if hypothetical) about how altruistic acts among hosts would benefit their microbes. It is easy to understand how increasing aggression (biting) in rabid mammals helps the rabies virus get to new hosts, but in this MS transmission to new hosts is built in to the model as one of the parameters. So in what additional ways might altruism among hosts help microbes?

We agree that this point was not clear enough in our original manuscript. We added an explanation of the role of interaction between horizontal and vertical transmission in the advantage of altruism (L44-51): "Following horizontal transfer, the microbes of the recipient host may become closely related to the microbes of the donating host, even when the two hosts are unrelated. Microbes can also transfer vertically, from parent to offspring. Thus, a microbe that induces its host to help another host increases the other host's survival or reproduction, and thus increases the vertical transmission of the microbes of the recipient host. Kin selection among the microbes could therefore favor microbes that induce altruistic behavior in their hosts, thereby increasing the vertical transmission of their microbial kin.", and some biological examples that may be relevant (L267-275).

It seems that every paper about altruism starts by stating what an unsolved problem altruism is in evolutionary biology, but the truth is that the theories mentioned by the authors: Kin selection, group selection, and reciprocal altruism have all been successful in explaining this phenomena. And they have all had many refinements since their introductory papers cited in the MS. The big controversy is not whether they work, but which is to be preferred in terms of simplicity or causal explanations-most of which really comes down to a preference for accounting methods. Their underlying similar mechanism is helping behaviors that are disproportionately directed towards the focal cooperative genotype. This assortment can be caused by conditional behavior (e.g. Tit-For-Tat), population viscosity, kin recognition, etc. The underlying similarities in these theories have been noted by many authors over the years, e.g., Breden 1990 *Evolution*, Wilson & Dugatkin 1997 *Am Nat*, Sober & Wilson 1998, Frank 1998, Page & Nowak 2002 *JTB*, Fletcher & Zwick 2006 *Am Nat*, although there has also been more recent criticisms of this equivalence viewpoint, e.g., van Veelen 2005 *JTB* and van Veelen et al 2012 *JTB*. The controversy about which theory is to be preferred should not be confused with failure of the theories to explain altruistic phenomena.

So I think it is too much to suggest that "microbes explain the evolution of host altruism", as if there is a void left by the failure of other theories that can be filled by microbe-

induced altruism. I think it would make more sense to say how microbe mediated altruism fits into the framework of more cooperative behaviors being directed to altruists.

We agree and thank the reviewer for pointing this out. First, we changed the description in the introduction and removed the “void”. Second, we added a new paragraph (L250-256) to the discussion where we explain the underlying similar mechanism of altruism preferentially directed towards altruists, and connect it to our model: “Our model can be considered in the context of classical theories for the evolution of altruism. It has been suggested that all previous models share a common driving force: that altruistic individuals preferentially help other altruistic individuals, according to kinship, memory, or group (see refs38, 39, 40, 41, 42, but see for a different view refs 43, 44). In our model the altruism-inducing microbe manipulates its host to help another host which might, after the interaction, carry the relatives of the original microbe – and thus help is preferentially directed towards future altruists. “

There is some attempt to connect to Hamilton's rule, but here there is the assumption that high relatedness automatically leads to altruism. The r term in Hamilton's rule is not simple genealogical relatedness, but rather a measure of above average covariance between the focal genotype and help from others.

We now clarify the similarity between our result and Hamilton's rule, and describe the latter more clearly (L104-120). We include a definition of relatedness, the lack of which contributed to the confusion in our original manuscript.

I also think the assumptions in the model are rather extreme. It is assumed that microbes have total control over their host's behavior, whereas in reality the influence of microbes on hosts is likely to be weak, especially if hosts can evolve defenses to having their behavior altered in ways that lower their fitness.

My biggest problem with the MS is that there does not seem to be any genetic basis in the model for hosts to be cooperators (independent of their microbes).

Also, while strong altruism may not evolve in well-mixed populations (without conditional behavior), it can come close to evolving if $b \gg c$. So microbial induced cooperation might push host cooperation to evolve when it is already close to evolving, but this enhancement of host altruism cannot be shown by the model if there is no host genetics for altruism. In other words, having an explicit trait for host altruism plus microbe caused altruism may give different results than assuming host altruism is zero and only looking at microbe caused altruism. Is there enhancement of host-mediated altruism by microbe-mediated altruism?

We thank the reviewer for pointing this out and helping us generalize the model we present. In our revised manuscript we relax this assumption in two ways:

- a. We allow for polymorphism with respect to altruism both in the host (alleles A and E, for an altruistic behavior with parameters b_g and c_g) and in the microbe (α and β , as before, with parameters b_m and c_m) in the same population. We thus have four types of hosts/individuals in the population. We study this generalized model (SI, section 1.5), and our results remain unchanged: genetic altruism does not evolve, even if it starts in complete linkage with microbe induced altruism. Microbe induced altruism evolves in exactly the same conditions independent of genetic altruism (SI, section 1.5). Even when considering $b_g \gg c_g$, microbe-induced altruism does not enhance the evolution of host altruism (Fig. S3). Intuitively, in the range where microbe-induced altruism evolves, horizontal transmission works a bit like recombination in classic genetic models, breaking the “advantageous” allele (microbe-induced altruism) from the “deleterious” background (host induced altruism).
- b. We now also investigate a case where there is a genetic baseline level of altruism among the hosts independent of the type of microbes. This is detailed in the general model section (SI section 1.1 and 1.2) and the results are presented in Fig. S2.

The MS seems to imply that there is: L17-18 "microbe-induced altruism can explain the evolution of altruism under wider conditions than host-centered theories" and L124-135 "transmission significantly extends the conditions allowing for the evolution and maintenance of altruism. But this cannot be assessed in the model if there is no

implementation of host-induced altruism.

We now present analysis (SI, section 1.6) and simulations (Fig 3,4) of altruism encoded in the host's genes. Throughout the description of the results we clarified the comparison between the host genetic model and the microbial one (Fig. 3, 'Gen' column and Fig. 4, star; main text L103-104, L190-191, L195-199, L201-204; SI 1.6-1.7).

We feel that this change strengthens our revised manuscript, and thank the reviewer for the very important comment.

I can understand how zero transmission could be used as a proxy for host genetics in the well mixed population, but this is confusing and at a minimum should be explained more clearly.

We admit that the presentation of the pure genetic case was confusing. It is now explained more clearly (L103-104, L190-191, L195-199, L201-204), and analyzed explicitly (see SI, section 1.6 and 1.7). We added to figures 3 and 4 an additional x-axis label showing the result of simulations done for the evolution of host allele for altruism.

Again, I think the idea of altruism being enhanced by microbes with their horizontal transfer is an interesting idea, but I don't think the current model, which seems to lack any explicit host genetics for altruistic traits, fully addresses this question.

We thank the reviewer again for helping us generalize the model as described above (see SI 1.1-1.3 and especially section 1.5) as well as better integrate the explicit host genetics into our work.

I also think some tie to nature would be very helpful. Under what circumstances might microbe-induced altruism between hosts be helpful to the microbes?

In our revised manuscript we explain the main advantage to the microbes from altruistic behavior: the vertical transmission of microbes to the offspring of the receiving host (L44-51).

We now better explain that inequality (1) in the main text is a condition regarding microbe fitness, thus providing an answer to under which conditions being an altruism-inducing microbe

is beneficial. We added to SI (section 1.4) an explicit derivation of the microbes' fitnesses and show how this derivation also results in inequality 1 from the main text. We also added more biological context in the introduction (L31-40) and discussion (L242-248, L267-275).

Specific Comments:

=====

L10-11: again, I don't agree that these theories have failed

We didn't have any intention to imply that these theories have failed, only that it is still a puzzle. We removed that sentence.

L40: again, natural selection would also favor hosts that could resist having their behaviors altered in ways that hurt their fitness.

This is very true. Coevolution is beyond the scope of this paper, but is clearly acknowledged (L258-260) and hopefully will be addressed in a further research.

L49: "altruism coded by the host's own genes" implies that there are such genes in the model, but if they are in the model they are not explained.

In our revised manuscript we investigate host-mediated altruism explicitly, with or without microbes (see derivations in SI sections 1.5 and 1.6). In the results we clarified the comparison between the host genetic model and the microbial one (Fig. 3, 'Gen' column and Fig. 4, star). See also relevant answers to previous comments about altruism encoded in the host genes.

L60-61: again "altruism encoded genetically in the host" suggest that such genes exist in the model.

See previous comments

L117-119: The idea of having repeated interactions (PD games) is that behavior in current interaction can be conditional on past interactions, e.g. Tit-For-Tat minimizes exploitation by defectors while gaining (mutual) benefits when interacting with other cooperative strategies. What is the point of repeated interactions in this model, if

behavior is not conditional?

As shown by Nowak and May 1992, altruism can spread in a population that inhabits a 2D lattice with 8 interactions per generation, even when behavior is not conditional. In our model, we got similar results (Fig. 3c, left column labeled 'Gen'). We found that when individuals in a structured population have 8 interactions per generation, altruism can spread in the population for some b/c values. But when individuals in a structured population have only one interaction per generation, altruism does not spread in the parameters studied, even for very high b/c values (Fig 3a and c). This is true even in the absence of transmission, and results from the spatial pattern of interaction: multiple interactions increase the competitive advantage of individuals with uniform payoff (e.g., altruistic individuals at the heart of a patch), in comparison with individuals with more variable payoff (e.g., egoistic individuals with mixed neighbors). Additionally, under repeated interactions the probability of recurrent interactions with the same individual increases significantly. Finally, in our model neighbors can become more "related" to each other microbially with each interaction due to horizontal transmission. All these characteristics contribute to the advantage of altruists with repeated interactions.

L125: why starting with such a high percent of alpha hosts?

Following the reviewer's comment, we investigated the dynamics starting from 5% rather than 50%, and the results are nearly unchanged. The results presented in the revised Fig. 3 start at 5% complementing the results in the revised figure 4 (ending at 5%). We believe presenting results starting at a low fraction of microbe-inducing microbes strengthens our work. Thanks for the comment!

L127: "As shown before" sounds like something discussed earlier in the MS, but I think authors are referring to other work in the literature. It might be clearer to say "As shown elsewhere" instead.

Changed

L129-132: This sentence sounds as if it is a finding from the model, but again where is

the host altruism genetics that could support this statement?

This is now detailed in the SI, section 1.6, and the results are shown explicitly in Fig 3 and Fig 4.

Fig. 3: In both a) and b) on the bottom row when $b/c = 1$ there are stable polymorphisms, but one would expect cooperators to go extinct when $b = c$ because even if $r = 1$, Hamilton's inequality is not satisfied. Can you explain what is going on here?

Note that condition (1) is different from Hamilton's rule in an important component: r is replaced by $(1 - T_\beta)/T_\alpha$. While r is always ≤ 1 , this does not have to be true for the ratio we study. For example, note that in the fully mixed analytical model the threshold for fixation reaches $b = c$ for $T = 0.5$ (when $T_\alpha = T_\beta$) and for lower values of T (when $T_\alpha > T_\beta$).

Consistent with the general advantages of spatial models for the evolution of altruism, the spatial model allows polymorphism even for lower (yet significant) transmission rates.

L151-153: This description of c) does not seem to match the legend on the y-axis which says "survival probability"

We revised Figure 3c (currently Figure 4) entirely: the y-axis label now reads: "prob (increase from rarity)" and is better explained in the figure caption (L230-236). We further changed the x axis to T parallel to fig 3 and fig 2 and included the genetic case explicitly (star).

L159-162: again, I don't think these examples are hard to explain using current theory.

Changed

REVIEWERS' COMMENTS:

Reviewer #2 (Remarks to the Author):

The authors have done an excellent job revising the manuscript after I previously reviewed it, and I thank them for their thorough and thoughtful responses to my comments. There are some minor remaining issues, which I list below, but in general the manuscript is very much improved.

2. Title: “microbe” is not wrong, but given that many microbes do not inhabit hosts, and the key point of this manuscript concerns microbes that do, I think it would be more informative to call them “symbionts” (especially here in the title, but also throughout the manuscript).

7. Replace “one that” with “which”

9. The explanation in this paper is based on kin selection (lines 13-14), so make clear that it’s not in addition to kin selection. Also, what do you mean by group selection (here and lines 26-27) – would multi-level selection be a better term?

16. Should be “microbe-induced host altruism” (or symbiont-induced, if you follow my earlier suggestion!)

17. Omit “for the first time”

18. Replace “recurring” with “repeated”

23. Omit the first sentence. The apparently obligatory initial reference to Darwin does not mean the subject is necessarily important! (Full disclosure: I’m guilty of this too.) And altruism has been studied before Darwin anyway!

25-26. Reword: the interactions aren’t direct/indirect – reciprocity is. Also, indirect reciprocity doesn’t require repeated interactions in the way that direct reciprocity does (because in indirect reciprocity, A helps B, B helps C, C helps A; no two players have to interact more than once, unlike in direct reciprocity).

32. Omit “microbes and”

42. Reword: “natural selection on microbes may favor manipulation of the host so that it acts altruistically” (in general the authors’ English is excellent; I’m a native speaker so I’m just making a few tweaks)

45. Replace “become” with “be”

47-51. Does this apply to horizontal transmission too? Horizontal transmission should also be facilitated by the host's survival and reproduction (because the longer the host lives and the more offspring it has, the more opportunities the microbe should have to be transmitted).

60-61. Reword: "representing the probabilities of microbes a and b being transmitted to and establishing in the other host, replacing..."

62. I'm not sure what you mean by "direct link" – is it that interaction is necessary for horizontal transmission to happen?

64. Replace "inherited" with "vertically transmitted" (relates better to earlier wording)

81. Reword: "Transmission and establishment of one microbe is independent of the other microbe." (Also line 48 in supplementary)

83-84. Replace "to interact" with "of interaction"

103-104. Specify "horizontal transmission of microbe a" – because here you still have $T_b \geq 0$, so microbe b can be horizontally transmitted.

108-109. Replace "can spread from rarity" with "is favored", because Hamilton's rule does not explicitly show the conditions under which alleles can spread

116-117. reword: "Thus, with probability T_a , manipulation by a microbes causes..."

123. In my previous review, I suggested that the authors rearrange Condition 1 in terms of b/c . I agree with their response explaining why they have Condition 1 in terms of Hamilton's rule, but I still think it would be really helpful, given how much the later text (and all three subsequent figures) is focused on this ratio. Keeping Condition 1 as is and presenting the b/c rearrangement afterwards would be ideal.

138. Change x-axis label to be "altruist microbe's transmission probability"

147. Omit "genetically" (redundant with "genome")

Fig 2. The figure itself is clear, but I found it hard to think through the verbal logic for what's going on, so maybe a few sentences in the caption or the text would be helpful. The main points I got are that 1) when c is higher, differences in horizontal and vertical transmission effectiveness have less of an effect; and 2) when $V_T < 1$, it is harder to evolve altruism when $T_b > T_a$ but easier when $T_a < T_b$. These results are not intuitive to me.

152. "Genetic altruism" is a misleading term, since microbe-induced altruism is also "genetic". How about something like "host-induced altruism" to contrast with microbe-induced?

155-156. Should be “the cost” and “the baseline level”

167. See comment on line 152: replace “genetically-encoded altruism” with “altruism encoded in the host’s genome” or similar

167-169. I wouldn’t say that consideration of these factors really relaxes the previous assumptions, which you deal with instead in the supplementary material. Here you’re relaxing the assumption of a fully mixed population, and adding additional assumptions relevant to spatial structure (e.g. interaction with immediate neighbors only).

183. Replace “his” with “its”

191. Replace with “host-induced altruism”

193. Specify analytical results “for a fully mixed population”

194-195. See comment for Fig 2: I don’t really understand the effect of imperfect vertical transmission, so a brief explanation would be helpful.

200. By “horizontal transmission probability” do you mean T_a or T_b or both?

203-205. I’m not quite sure what you mean here. Host-genome-encoded altruism could still have imperfect vertical transmission, e.g. if there were a high mutation rate.

219. Should be “the y-axis”

223. Add hyphen in “microbe-induced”

224-225. It seems like you’ve changed two things here compared to the preceding model. First, as you state, the altruist microbe is rare (starting now at 0.0004 compared to 0.05 previously), but second, you also have a non-random distribution (2x2 patch, instead of randomly distributed). Shouldn’t one of these factors be kept the same?

230. Should be “extremely”

233. I assume that $T_a=T_b=T$ here (as for Fig 3)?

236. Should be “measured”

237. Reword: “the probability that a increases”

239. Reword: “the probability that a survives”

240. Is “a neutral microbe” the same as microbe b?

240-241. Reword: “the probability that altruists increase in proportion from rarity when...”

243-244. I think this is an overstated claim. I agree that the model here gives intriguing results, and does indeed have interesting implications for the evolution of altruism without spatial structure, repeated interactions or recognition (lines 248-249), but it does not “suggest a new perspective on

almost any manifestation of altruistic behavior". The examples presented in lines 245-248 are all well explained by existing hypotheses, which is not to say that new hypotheses cannot or do not also apply, but the authors do not show that there is even any preliminary evidence that individuals in these examples exchange microbes. The citations here also seem oddly chosen: for example, ref 36 is not really about mutualism in the sense of inter-specific cooperation (for the large literature on cooperation in inter-specific mutualism, see e.g. Bronstein 2016 "Mutualism", Oxford University Press), and the authors should be aware of the multiple responses to reference 33 (e.g. Abbot et al. 2011 Nature; better to cite instead something like Bourke 2011 Proc R Soc B).

252-253. I don't think this is strictly true: as long as altruism allows the altruist to ultimately gain inclusive fitness benefits, then it does not matter whether the recipients are also altruists. For example, a subordinate wasp helps the dominant because doing so increases the chance of direct benefits of nest inheritance (Leadbeater et al. 2011 Science), and donating to charity gives the opportunity to signal quality in male-male competition for access to mates (Raihani & Smith 2015 Curr Biol).

254-257. I don't understand the point about help being preferentially directed to future altruists. The act of altruism only helps the altruism allele if 1) the recipient host already has it, or 2) the donor host transmits it to the recipient (with probability T_a). In the model there is no preferential interaction with hosts that already carry the altruism allele (by the authors' own admission, e.g. next paragraph, and elsewhere where they cite the lack of recognition as a unique feature of the model). Altruism is indeed more likely to evolve in this model when T_a is higher and the altruism allele is more likely to be transmitted; if this is what is meant by preferential direction of help, it should be reworded.

260-261. "a co-evolutionary arms race with respect to altruistic behavior" sounds like there is an arms race to be more altruistic. An extra sentence to explain what you mean here would be useful.

265-266. I'm not sure what you mean here – again, an extra sentence to elaborate would be helpful.

268-269. This is a very good point and suggests many opportunities for empirical tests of the model, which I think would be worth mentioning briefly here, e.g. to see whether treatment of hosts with antibiotics or probiotics decreases or increases altruistic behavior.

270. Replace "can" with "could", and reword "indirect" – I'm not sure what it means in this context

References: please check journal titles – not all of them are properly capitalized

Supplementary sections 1.1, 1.2, 1.3: My background is not in population genetics, so I don't follow all of the derivations, but the logic seems sound, and I thank the authors for providing these new details.

50-51. Change "his" to "its" (asexual)

57. Change to "definition of parameters". In this section I would use colons rather than hyphens, since it's a little confusing with the minus signs (also in Section 1.4, 243-249).

58-61. I find the descriptions "microbial c, b " and "genetic c, b " for cost and benefits a little confusing: aren't these the costs and benefits of host altruism and of microbial altruism? This comes up again on lines 77-79: c_g is the cost of altruism encoded by the host's genes, so why should this be changed when the cost of carrying the microbe changes?

68. What does ":= " mean?

88. Replace "get" with "find"

Lines 123 (if $\mu < T_a/T_b$, S_6 is never satisfied...) and 132 (for $\mu < T_a/T_b$, S_6 is satisfied...) seem to contradict each other

Section 1.4: Thank you – I now follow your calculations of relatedness!

254. State that each player has a baseline fitness of 1?

260. Should be "selfish individual"

Sections 1.5, 1.6, 1.7: Again, I can't really assess this because I don't follow all of the population genetics, but superficially the logic seems solid.

301. Would it be better to use different variables? Keep p and q the same as before and make the frequency of each type a factor?

302, 304. I don't follow these steps – are they based on the same method as section 1.1?

337. Delete comma

339. Replace ; with (

341. "nor" should be "or"

Section 2: Thank you for providing the simulation workflow and stopping criteria – this is all much clearer now!

382. Replace "drawn at" with "drawn in"

404. Replace “as stable” with “to be stable”

418. Replace “get” with “find”

421-422. Replace “more significant” with “larger”. I don’t immediately see why this is the case (i.e. why the horizontal transmission advantage matters more when there are more interactions per generation), so a very brief explanation, if you have one, would be interesting.

Reviewer #3 (Remarks to the Author):

The MS has been updated and markedly improved by the authors. Claims have been toned down, models have been enhanced and many additional simulations have been run.

That said, I think L 243-244 still claim too much, especially given that this model has some important limitations. To me this MS is interesting because it shows theoretically that if microbe induced altruism existed it could contribute to the explanation of altruism in nature, and maybe even make it easier to understand how it could arise. But no empirical evidence is offered that such microbe induced altruism actually exists. Also, in terms of model assumptions, the idea that microbes would, through host interactions, easily take over other hosts (that already have their own microbes), seems unrealistic. My knowledge of microbiome research is scant, but one aspect I have heard researchers talk about is the stability of these microbe communities. So when you shake someone’s hand some microbes are exchanged both ways, but shortly afterwards the microbiomes in each hand are returned to their original state. This may mean that horizontal transmission would be quite limited. Lastly, the idea that hosts would so easily surrender their behaviors to the will of their microbes also seems unrealistic. So I think this model is a “best case” for the influence of microbes on host altruistic behavior, and it is therefore way too early to claim that all manifestations of altruism need to be rethought.

I do think Figure 3 is especially nice in comparing host and microbe mediated altruism when acting separately (again, given the favorable assumptions of the model). In the future, it would be interesting to explore how the two types interact. Am I correct that the “Gen” column shows final proportion of A (rather than alpha) and that the adjacent plots show runs with microbe altruism turned on, but host-mediated altruism turned off? There does seem to be an attempt to look at the two together in the SI 1.5, but here in the analytical model host altruism seems to be doomed (SI 1.6 and 1.7). Were the four types (A-alpha, E-alpha, A-beta, and E-beta) explored in the spatial model? I wasn’t clear on this.

Specific Comments:

=====

L 25: it is a common misconception that kinship automatically favors altruism among kin. Consider changing “favors” to “can favor”

L 63-65: It might be useful to note here that generations are discrete—parent generation all die.

L 80-81: Does order of operations affect the outcome? That is, does it matter that alphas replace betas first and then betas replace alphas?

Fig. 3: If color shades represent final proportion of A rather than alpha in the case of host-mediated altruism, saying this explicitly would make it more clear.

L 211-212: I found the labelling of subgraph parameters confusing at first. Isn't it more conventional to put the subgraph label, e.g. (a), before the list of parameters for that graph?

L 225: Neither of the videos showed much when I played them. Neither changed at all after the first couple of seconds until the end.

L 226: I didn't understand why $p = 0.05$ was used as the cutoff.

SI L 380: In most spatial models that I am familiar with, edge effects are eliminated by treating a 2-D grid as a toroidal. Is there a reason for not doing this here? What affect do edges have on the results?

REVIEWERS' COMMENTS:

Reviewer #2 (Remarks to the Author):

The authors have done an excellent job revising the manuscript after I previously reviewed it, and I thank them for their thorough and thoughtful responses to my comments. There are some minor remaining issues, which I list below, but in general the manuscript is very much improved.

2. Title: “microbe” is not wrong, but given that many microbes do not inhabit hosts, and the key point of this manuscript concerns microbes that do, I think it would be more informative to call them “symbionts” (especially here in the title, but also throughout the manuscript).

We indeed debated a lot about this point, finding pros and cons to each option. We eventually decided to use “microbe” since we think it better reflects the fact that the entity we consider is one which manipulates its host to act against its immediate good. We believe that “symbionts” misses this distinction. We do, however, mention the extension to symbionts in general in the introduction: “Almost any organism hosts microbes or other symbionts. A growing body of

evidence shows that microbes and symbionts can mediate behavioral changes in their hosts, in some cases improving their own fitness and transmission ability". (L33).

7. Replace "one that" with "which"

Thanks. Replaced. (L13)

9. The explanation in this paper is based on kin selection (lines 13-14), so make clear that it's not in addition to kin selection. Also, what do you mean by group selection (here and lines 26-27) – would multi-level selection be a better term?

We rephrased the abstract and hope that this confusion is now solved.

Both group selection and multi-level selection are suitable terms. We chose "group selection" simply because it is more consistent with our ref (Wilson DS. A theory of group selection. Proceedings of the National Academy of Sciences 72, 143-146 (1975)).

16. Should be "microbe-induced host altruism" (or symbiont-induced, if you follow my earlier suggestion!)

Thanks. Changed throughout the paper. (L18)

17. Omit "for the first time"

Thanks. Omitted.

18. Replace “recurring” with “repeated”

Thanks. Changed.

23. Omit the first sentence. The apparently obligatory initial reference to Darwin does not mean the subject is necessarily important! (Full disclosure: I’m guilty of this too.) And altruism has been studied before Darwin anyway!

We agree that the reference to Darwin is sometimes overused, but our focus here is not merely on altruism, which was indeed studied before Darwin, but rather on the *evolution* of altruism which has first been discussed by Darwin to the best of our knowledge. Thus we prefer to keep this first sentence.

25-26. Reword: the interactions aren’t direct/indirect – reciprocity is. Also, indirect reciprocity doesn’t require repeated interactions in the way that direct reciprocity does (because in indirect reciprocity, A helps B, B helps C, C helps A; no two players have to interact more than once, unlike in direct reciprocity).

Thanks. Indeed this sentence was not clear. We changed to “...reciprocity, which suggests repeating interactions or individual recognition, as key factors...”. (L27)

32. Omit “microbes and”

Since we remained with “microbes” (see our response above), we haven’t changed the sentence.

42. Reword: “natural selection on microbes may favor manipulation of the host so that it acts altruistically” (in general the authors’ English is excellent; I’m a native speaker so I’m just making a few tweaks)

Thanks. Reworded. (L44)

45. Replace “become” with “be”

We rephrased the sentence to “Following horizontal transfer, the recipient host may carry microbes that are closely related to the microbes of the donating host, even when the two hosts are unrelated.” (L47)

47-51. Does this apply to horizontal transmission too? Horizontal transmission should also be facilitated by the host’s survival and reproduction (because the longer the host lives and the more offspring it has, the more opportunities the microbe should have to be transmitted.

We removed “vertical” in our description, so the sentence now reads “Kin selection among the microbes could therefore favor microbes that induce altruistic behavior in their hosts, thereby increasing the transmission of their microbial kin” (L51). We do note that in the model (which is described later in the paper) the fitness determines only reproductive success and not longevity (generation time is fixed in the population, and there is only one interaction per generation).

Therefore, only the fact that the more offspring the host has, the more opportunities the microbe has to be transmitted is reflected in our model. Nevertheless, the introduction describes the general model, which may apply also to horizontal transmission. Thanks.

60-61. Reword: “representing the probabilities of microbes a and b being transmitted to and establishing in the other host, replacing...”

Thanks. Reworded to “During host interaction, microbes can be transmitted between the interacting hosts with probabilities T_α and T_β . T_α represents the probability of microbes of type α being transmitted to the other host, replacing the resident microbes, and likewise for T_β ” (L63).

62. I’m not sure what you mean by “direct link” – is it that interaction is necessary for horizontal transmission to happen?

Indeed. This sentence is meant to stress that interaction is coupled with a chance for horizontal transmission, and that interaction (in which individuals help or defect) is the act in which microbes can be transferred. We rephrased the sentence to better clarify this, and we hope that we now explain it better: “This direct link between interaction and the possibility for horizontal transmission is at the core of our model and differs from all related works”. (L66)

64. Replace “inherited” with “vertically transmitted” (relates better to earlier wording)

Thanks. Replaced. (L68)

81. Reword: “Transmission and establishment of one microbe is independent of the other microbe.” (Also line 48 in supplementary)

Thanks. We rephrased to: “Transmission and establishment of one microbe is independent of the other microbe, and when both occur, they occur simultaneously.” (L598)

83-84. Replace “to interact” with “of interaction”

Thanks. Replaced. (L79)

103-104. Specify “horizontal transmission of microbe a” – because here you still have $T_b \geq 0$, so microbe b can be horizontally transmitted.

Thanks. Done. (L99)

108-109. Replace “can spread from rarity” with “is favored”, because Hamilton’s rule does not explicitly show the conditions under which alleles can spread

Thanks. Replaced. (L104)

116-117. reword: “Thus, with probability T_a , manipulation by a microbes causes...”

Thanks. Reworded. (L111)

123. In my previous review, I suggested that the authors rearrange Condition 1 in terms of b/c. I agree with their response explaining why they have Condition 1 in terms of Hamilton’s rule, but I still think it would be really helpful, given how much the later text (and all three subsequent figures) is focused on this ratio. Keeping Condition 1 as is and presenting the b/c rearrangement afterwards would be ideal.

This is a good solution – we changed this in the text as follows: “Under equal horizontal transmission $T_\alpha = T_\beta = T$, condition (1) reduces to a simpler form $\frac{b}{c} > \frac{1-T}{T}$.” (L90), and in the caption of figure 2 as follows: “When the horizontal transmission probabilities are equal $T_\alpha = T_\beta = T$ (green solid lines), the condition for the spread of altruism becomes $\frac{b}{c} > \frac{1-T}{T}$, for any $VT > 0$...”. (L605)

138. Change x-axis label to be “altruist microbe’s transmission probability”

Thanks. We changed to “Transmission probability of the altruism-inducing microbe (T_α)”.

147. Omit “genetically” (redundant with “genome”)

Thanks. Omitted.

Fig 2. The figure itself is clear, but I found it hard to think through the verbal logic for what’s going on, so maybe a few sentences in the caption or the text would be helpful. The main points I got are that 1) when c is higher, differences in horizontal and vertical transmission effectiveness have less of an effect; and 2) when $VT < 1$, it is harder to evolve altruism when $T_b > T_a$ but easier when $T_a < T_b$. These results are not intuitive to me.

Thanks! We added the following paragraph, which we hope explains it better, to the figure caption: “Thus, the line depends only on T and is identical in all three subplots. However, the altruism inducing bacteria spreads more slowly when $VT < 1$ (see Supplementary Note 1.5). As c increases (from Fig. 2a to Fig. 2c), the fitness effect of interaction on vertical transmission increases, diminishing the relative effect of imbalance between the horizontal transmission

rates. The effect of imperfect vertical transmission ($VT < 1$), is opposite, diminishing the effect of fitness differences on vertical transmission, thus giving more weight to imbalance between the horizontal transmission rates (compare red and blue solid lines to dashed lines). “ (L607) We further added a new analysis in Supplementary Note 1.5, investigating the effect of VT on the rate of spread of altruism inducing microbe.

152. “Genetic altruism” is a misleading term, since microbe-induced altruism is also “genetic”.

How about something like “host-induced altruism” to contrast with microbe-induced?

Good point. Thanks. We changed this and other sentences to avoid using the term “Genetic altruism”.

155-156. Should be “the cost” and “the baseline level”

Thanks. This sentence is now revised.

167. See comment on line 152: replace “genetically-encoded altruism” with “altruism encoded in the host’s genome” or similar

Thanks. Changed. (L151)

167-169. I wouldn’t say that consideration of these factors really relaxes the previous assumptions, which you deal with instead in the supplementary material. Here you’re relaxing the assumption of a fully mixed population, and adding additional assumptions relevant to spatial structure (e.g. interaction with immediate neighbors only).

Indeed. Thanks. We completely rephrased the sentence: “By studying spatial models, we extend our analysis to populations that are subject to drift, local interactions, local transmissions, and limited dispersal.” (L151)

183. Replace “his” with “its”

Thanks. Replaced. (L167)

191. Replace with “host-induced altruism”

Thanks. We replaced it with “altruism encoded in the host’s genome”. (L175)

193. Specify analytical results “for a fully mixed population”

Thanks. Added. (L177)

194-195. See comment for Fig 2: I don’t really understand the effect of imperfect vertical transmission, so a brief explanation would be helpful.

We added a new part in the caption of figure 2, and expanded the current part with the following sentence: “Assuming that the vertical transmission of microbes is imperfect ($VT < 1$) somewhat narrows the parameter range allowing the evolution of microbe-induced altruism, since it reduces the advantage of altruism-inducing microbes, which is based on enhancing the vertical transmission of the microbes in the recipient host (Fig. 3b)” (L178). We hope these additions help in clarifying this point.

200. By “horizontal transmission probability” do you mean T_α or T_β or both?

Here the horizontal transmission probability is the same for both microbes, $T_\alpha = T_\beta = T$. We clarified this point in the text to avoid confusion:

“...but the parameter range allowing persistence is wider for microbe-induced altruism, and widens with horizontal transmission probability ($T = T_\alpha = T_\beta$) (Fig. 3c)” (L185)

203-205. I’m not quite sure what you mean here. Host-genome-encoded altruism could still have imperfect vertical transmission, e.g. if there were a high mutation rate.

This is true. In order to clarify that we are neglecting mutations we modified the following sentence in the model description: “We compare the evolution of microbe-induced altruism with the classical case of altruism encoded genetically in the host, with perfect vertical transmission, no horizontal transmission, neglecting mutations, and using the same parameters b and c . ” (L74)

219. Should be “the y-axis”

Thanks. Done. (L631)

223. Add hyphen in “microbe-induced”

Thanks. Done. (L194)

224-225. It seems like you’ve changed two things here compared to the preceding model. First, as you state, the altruist microbe is rare (starting now at 0.0004 compared to 0.05 previously),

but second, you also have a non-random distribution (2x2 patch, instead of randomly distributed). Shouldn't one of these factors be kept the same?

An approach consistent with the above would have been to start the simulations of initial proportion of 5% with the results of the rare patch, or in other words to continue the simulation starting from a rare patch all the way to a stabilized result. The issue here is that those simulations are very stochastic and harder to reproduce. We thus thought that separating the question into two parts, namely (1) increase from a rare patch to 5% and (2) stabilization in the population starting from 5% distributed *randomly*, would allow the generation of a very large and reproducible set of results in a reasonable time. We do acknowledge that this is not the only way to design these simulations, but we feel this does not have a real effect on the strength of the results.

230. Should be "extremely"

Thanks. Fixed. (L635)

233. I assume that $T_a = T_b = T$ here (as for Fig 3)?

Indeed, thanks. Changed to: "...is plotted as a function of $T = T_\alpha = T_\beta$ ". (L638)

236. Should be "measured"

Thanks. Fixed. (L641)

237. Reword: "the probability that a increases"

Thanks. In order to avoid using “increase” twice, we changed to: “This estimated probability that α will increase grows with T”. (L642)

239. Reword: “the probability that a survives”

Thanks. Reworded. (L643)

240. Is “a neutral microbe” the same as microbe b?

Thanks for pointing this out. We added clarification as follows:

“The probability that α survives, when $T=0$, was found not to be higher than 4/10000 based on 50,000 runs (the probability of a neutral microbe, identical in its effect on behavior to microbe β , to fixate in such a model).” (L643)

240-241. Reword: “the probability that altruists increase in proportion from rarity when...”

This sentence is now changed to “The star (“Gen”) represents the case of altruistic behavior encoded in the host genome, where altruists do not increase from rarity for any b/c value” (L645)

243-244. I think this is an overstated claim. I agree that the model here gives intriguing results, and does indeed have interesting implications for the evolution of altruism without spatial structure, repeated interactions or recognition (lines 248-249), but it does not “suggest a new perspective on almost any manifestation of altruistic behavior”. The examples presented in lines 245-248 are all well explained by existing hypotheses, which is not to say that new

hypotheses cannot or do not also apply, but the authors do not show that there is even any preliminary evidence that individuals in these examples exchange microbes. The citations here also seem oddly chosen: for example, ref 36 is not really about mutualism in the sense of inter-specific cooperation (for the large literature on cooperation in inter-specific mutualism, see e.g. Bronstein 2016 "Mutualism", Oxford University Press), and the authors should be aware of the multiple responses to reference 33 (e.g. Abbot et al. 2011 Nature; better to cite instead something like Bourke 2011 Proc R Soc B).

We agree with this comment. First, we rephrased the sentence to now read: "...imply a new perspective on various manifestations of altruistic behavior." (L203). Second, we replaced the refs according to the reviewer's suggestion. Thanks!

252-253. I don't think this is strictly true: as long as altruism allows the altruist to ultimately gain inclusive fitness benefits, then it does not matter whether the recipients are also altruists. For example, a subordinate wasp helps the dominant because doing so increases the chance of direct benefits of nest inheritance (Leadbeater et al. 2011 Science), and donating to charity gives the opportunity to signal quality in male-male competition for access to mates (Raihani & Smith 2015 Curr Biol).

Thanks. We rephrased the sentence, which now reads: "It has been suggested that many of the previous models share a common principle" (L210). Note that our model considers only cases of "strong" altruism - behavior that directly reduces the fitness of its performer.

254-257. I don't understand the point about help being preferentially directed to future

altruists. The act of altruism only helps the altruism allele if 1) the recipient host already has it, or 2) the donor host transmits it to the recipient (with probability T_a). In the model there is no preferential interaction with hosts that already carry the altruism allele (by the authors' own admission, e.g. next paragraph, and elsewhere where they cite the lack of recognition as a unique feature of the model). Altruism is indeed more likely to evolve in this model when T_a is higher and the altruism allele is more likely to be transmitted; if this is what is meant by preferential direction of help, it should be reworded.

This is indeed a point that needs better clarification. We rephrased this part to now read:

“In our model the altruism-inducing microbe manipulates its host to help another host, irrespective of its microbes. Following the interaction, the receiving host may carry the relatives of the original microbe, and thus help is in effect preferentially directed towards future altruists. That is, the probability of helping someone that would be an altruist after the interaction ($p+qT_\alpha$) is higher than the proportion of altruists in the general population (p) “.

(L213)

260-261. “a co-evolutionary arms race with respect to altruistic behavior” sounds like there is an arms race to be more altruistic. An extra sentence to explain what you mean here would be useful.

Thanks. We changed the sentence that now reads:

“Such a conflict can lead to a co-evolutionary arms race with respect to altruistic behavior, where the host evolves resistance to the altruism-inducing microbes, and the microbes evolve new ways of manipulating the host.” (L220)

265-266. I'm not sure what you mean here – again, an extra sentence to elaborate would be helpful.

Thanks again for helping us locate unclear points. We rephrased it to now read:

“Third, more realistic modeling of the host microbiome could consider a diverse microbial population within a single host, where behavior is determined by microbial composition.”

(L226)

268-269. This is a very good point and suggests many opportunities for empirical tests of the model, which I think would be worth mentioning briefly here, e.g. to see whether treatment of hosts with antibiotics or probiotics decreases or increases altruistic behavior.

The last sentence of the manuscript has been rephrased, and now refers specifically to empirical tests and antibiotics: “Our theoretical predictions call for experimental validation of whether microbes indeed mediate altruistic behavior of their hosts, by what mechanisms, and whether elimination of microbes, e.g. by antibiotics, hampers altruism.” (L238)

270. Replace “can” with “could”, and reword “indirect” – I'm not sure what it means in this context

Thanks. We replaced the “can” with “could”, and added an explanation for the indirect effect:

“In many cases the effect on altruistic behavior could be an indirect result of an effect on other behaviors: for example reduction of social anxiety...” (L230)

References: please check journal titles – not all of them are properly capitalized

Thanks. Done.

Supplementary sections 1.1, 1.2, 1.3: My background is not in population genetics, so I don't follow all of the derivations, but the logic seems sound, and I thank the authors for providing these new details.

50-51. Change "his" to "its" (asexual)

Done (section moved to methods).

57. Change to "definition of parameters". In this section I would use colons rather than hyphens, since it's a little confusing with the minus signs (also in Section 1.4, 243-249).

We moved this part to methods and the parameters are not introduced in a list anymore.

58-61. I find the descriptions "microbial c, b" and "genetic c, b" for cost and benefits a little confusing: aren't these the costs and benefits of host altruism and of microbial altruism? This comes up again on lines 77-79: c_g is the cost of altruism encoded by the host's genes, so why should this be changed when the cost of carrying the microbe changes?

We clarified our description, which now reads (section moved to methods): "Note that this formulation also covers the case where there is an intrinsic cost to carrying a microbe: Since c_g is uniform across the population, an equal cost for all microbe types can be introduced through an increase in c_g . Different costs to the different microbe types can be introduced by changing

c (and assuming the cost to carrying the microbe is applied before any horizontal transfer occurs).” (L257)

68. What does “:=” mean?

This symbol a definition. We now refrain from using the symbol and clarify the definitions verbally.

88. Replace “get” with “find”

Done (section moved to methods).

Lines 123 (if $\mu < T_a/T_b$, S_6 is never satisfied...) and 132 (for $\mu < T_a/T_b$, S_6 is satisfied...) seem to contradict each other

Thanks! This was indeed a typo, and this section was unclear. We revised it (Supplementary Note 1.1).

Section 1.4: Thank you – I now follow your calculations of relatedness!

254. State that each player has a baseline fitness of 1?

Thanks. Stated.

260. Should be “selfish individual”

Thanks. Done.

Sections 1.5, 1.6, 1.7: Again, I can't really assess this because I don't follow all of the population genetics, but superficially the logic seems solid.

301. Would it be better to use different variables? Keep p and q the same as before and make the frequency of each type a factor?

Changed according to the reviewer's suggestion. Thanks!

302, 304. I don't follow these steps – are they based on the same method as section 1.1?

Yes! we now mention that (sup line 236).

337. Delete comma

Thanks. Done (section rewritten in the methods)

339. Replace ; with (

Thanks. Done.

341. "nor" should be "or"

Thanks. Done.

Section 2: Thank you for providing the simulation workflow and stopping criteria – this is all much clearer now!

382. Replace “drawn at” with “drawn in”

Thanks. Done.

404. Replace “as stable” with “to be stable”

Thanks. Done.

418. Replace “get” with “find”

Thanks. Done.

421-422. Replace “more significant” with “larger”. I don’t immediately see why this is the case (i.e. why the horizontal transmission advantage matters more when there are more interactions per generation), so a very brief explanation, if you have one, would be interesting.

We changed to “larger” and added an explanation: “When we increase the number of interactions to $K=8$, the same change in horizontal transmission ratio has a somewhat larger effect, as the same rate of horizontal transmission is applied 8 times in each generation. “ (Sup L281)

Reviewer #3 (Remarks to the Author):

The MS has been updated and markedly improved by the authors. Claims have been toned down, models have been enhanced and many additional simulations have been run.

That said, I think L 243-244 still claim too much, especially given that this model has some important limitations. To me this MS is interesting because it shows theoretically that if microbe induced altruism existed it could contribute to the explanation of altruism in nature, and maybe even make it easier to understand how it could arise. But no empirical evidence is offered that such microbe induced altruism actually exists. Also, in terms of model assumptions, the idea that microbes would, through host interactions, easily take over other hosts (that already have their own microbes), seems unrealistic. My knowledge of microbiome research is scant, but one aspect I have heard researchers talk about is the stability of these microbe communities. So when you shake someone's hand some microbes are exchanged both ways, but shortly afterwards the microbiomes in each hand are returned to their original state. This may mean that horizontal transmission would be quite limited.

Lastly, the idea that hosts would so easily surrender their behaviors to the will of their microbes also seems unrealistic. So I think this model is a "best case" for the influence of microbes on host altruistic behavior, and it is therefore way too early to claim that all manifestations of altruism need to be rethought.

We accept this comment and have toned down these lines accordingly, to now read: "Our results – that microbes can facilitate the evolution of host altruism – imply a new perspective on various manifestations of altruistic behavior". (L203)

I do think Figure 3 is especially nice in comparing host and microbe mediated altruism when acting separately (again, given the favorable assumptions of the model). In the future, it would

be interesting to explore how the two types interact. Am I correct that the “Gen” column shows final proportion of A (rather than alpha) and that the adjacent plots show runs with microbe altruism turned on, but host-mediated altruism turned off? There does seem to be an attempt to look at the two together in the SI 1.5, but here in the analytical model host altruism seems to be doomed (SI 1.6 and 1.7). Were the four types (A-alpha, E-alpha, A-beta, and E-beta) explored in the spatial model? I wasn’t clear on this.

We thank the reviewer for pointing out that this figure was not described clearly enough. We revised the legend of figure 3 and we hope it better clarifies what is depicted in the figure. The case of 4 types in a spatial simulation is an interesting future direction.

Specific Comments:

=====

L 25: it is a common misconception that kinship automatically favors altruism among kin.

Consider changing “favors” to “can favor”

Thanks. Changed. (L27)

L 63-65: It might be useful to note here that generations are discrete—parent generation all die.

Thanks. We now describe that assumption and the sentence now reads:

“At the end of each generation, individuals reproduce according to their fitness, microbes are vertically transmitted from one generation to the next, and the offspring generation replaces the parent generation.” (L67)

L 80-81: Does order of operations affect the outcome? That is, does it matter that alphas replace betas first and then betas replace alphas?

Thanks. We now explain that in the figure caption: “Transmission and establishment of one microbe is independent of the other microbe, and when both occur, they occur simultaneously.” (L598)

Fig. 3: If color shades represent final proportion of A rather than alpha in the case of host-mediated altruism, saying this explicitly would make it more clear.

Again, we thank the reviewer for pointing out that figure 3 was not explained well enough. As we stated above, we revised the figure legend and we now state this clearly.

L 211-212: I found the labelling of subgraph parameters confusing at first. Isn't it more conventional to put the subgraph label, e.g. (a), before the list of parameters for that graph?

Thanks for pointing this out. We changed so that the subgraph labels are before the list of parameters.

L 225: Neither of the videos showed much when I played them. Neither changed at all after the first couple of seconds until the end.

We uploaded the files again, and verified that they open from our computer. Please let us know if there are still problems.

L 226: I didn't understand why $p = 0.05$ was used as the cutoff.

From fig3 we already knew what happens to the altruists when they start from 5%. So in fig4 we just wanted to analyze the chance that a rare patch survives and reaches 5%.

Technically, stopping the simulation at 5% (instead of letting the simulation end when the population stabilizes) allowed us run much more simulations.

SI L 380: In most spatial models that I am familiar with, edge effects are eliminated by treating a 2-D grid as a toroidal. Is there a reason for not doing this here? What affect do edges have on the results?

We debated about this point. Torus indeed eliminates edge effects but it has its own artifacts (distant parts of the population are more connected than expected otherwise). Also, natural populations usually don't live on a torus, and therefore some prefer to avoid that. Since this is a debatable issue, we decided to follow a previous work (Nowak et. Al. 1992) and remain with edges. Furthermore, in our simulation, we normalize the fitness of each individual by the number of interactions it had to diminish the edge effect without changing the level of connectivity in the population.